# RELATIONAL LEARNING WITH VARIATIONAL BAYES

**Kuang-Hung Liu**[*]
ExxonMobil Research and Engineering
`kuang-hung.liu@exxonmobil.com`

## ABSTRACT

In psychology, relational learning refers to the ability to recognize and respond to relationship among objects irrespective of the nature of those objects. Relational learning has long been recognized as a hallmark of human cognition and a key question in artificial intelligence research. In this work, we propose an unsupervised learning method for addressing the relational learning problem where we learn the underlying relationship between a pair of data irrespective of the nature of those data. The central idea of the proposed method is to encapsulate the relational learning problem with a probabilistic graphical model in which we perform inference to learn about data relationship and other relational processing tasks.

## 1 INTRODUCTION

American Psychological Association defines *relational learning* as (VandenBos & APA, 2007):

**Definition 1.1** (**Relational learning**). *Learning to differentiate among stimuli on the basis of relational properties rather than absolute properties.*

In other words, relational learning refers to the ability to recognize and respond to relationship (called *relational property*) among objects *irrespective* of the nature of those objects (called *absolute property*). For example (attributed to Doumas & Hummel (2013)), how do we come to understand that two circles are the *same-shape* in the same way that two squares are? In this example, "same-shape" is the relational property and object shape is the absolute property. Relational learning has long been recognized as a hallmark of human cognition with strong implications for both human-like learning capabilities and generalization capacity (Biederman, 1987; Medin et al., 1993; Gentner, 2003; Penn et al., 2008; Holyoak, 2012; Gentner & Smith, 2012). We refer the interested readers to the provided references for a comprehensive discussion on this subject. Contemporaneously, the research on learning data relationships—also commonly called "relational learning"—has flourished in the machine learning community where the overarching goal is learning in a context where there may be relationships between learning examples, or where these examples may have a complex internal structure (i.e., consist of multiple components and there may be relationships between these components) (Getoor & Taskar, 2007; De Raedt et al., 2016). We argue that the key difference between the two "relational learning" definitions and their learning objectives is that Definition 1.1 takes the relationship learning problem one step further by requiring the data relationships be learned only on the basis of relational properties rather than absolute properties. To the best of our knowledge, this important distinction—learning relationships irrespective of the absolute properties—has not been rigorously studied in the unsupervised learning community, where most existing methods either encourage or do not constrain the relationships learning through absolute properties.

In this work, we propose an unsupervised learning method—variational relational learning (VRL)—for addressing the relational learning problem as defined by Definition 1.1. At its core, VRL encapsulates the relational learning problem with a probabilistic graphical model (PGM) in which we perform inference to learn about relational property and other relational processing tasks. Our contribution in this paper is threefold: First, we propose a probabilistic formulation for the relational learning problem defined by Definition 1.1. Second, we encapsulate the relational learning problem with a PGM in which we perform learning and inference. Third, we propose an efficient and effective learning algorithm that can be trained end-to-end and completely unsupervised.

---

[*]The author completed this research while working at ExxonMobil. Corresponding author e-mail: `liu.kuanghung@gmail.com`

## 2 PROBLEM DEFINITION

We focus on a canonical form of the relational learning problem where we observed a paired dataset $\mathbf{X} = \{\, (\mathbf{a}^{(i)}, \mathbf{b}^{(i)}) \mid i \in [1..N] \,\}$ consisting of $N$ *i.i.d.* samples generated from a joint distribution $p(\,\mathbf{a} \in \mathcal{A}, \mathbf{b} \in \mathcal{B}\,)$. We dissect the information in $\mathbf{X}$ into *absolute property* and *relational property* where absolute property represents specific features that describe individual $\mathbf{a}$ and $\mathbf{b}$, and relational property represents the relationship between $\mathbf{a}$ and $\mathbf{b}$ *irrespective* of their absolute property. In this work, we interpret the *absolute property* of $\mathbf{a}$ and $\mathbf{b}$ as any information that characterizes (even if only partially) the marginal distribution $p(\,\mathbf{a}\,)$ and $p(\,\mathbf{b}\,)$. We propose to represent the *relational property* as a latent random variable (r.v.) $\mathbf{z}$ that satisfies the following constraints:

$$\text{(i) } p(\,\mathbf{a}, \mathbf{z}\,) = p(\,\mathbf{a}\,)p(\,\mathbf{z}\,), \quad \text{(ii) } p(\,\mathbf{b}, \mathbf{z}\,) = p(\,\mathbf{b}\,)p(\,\mathbf{z}\,),$$

$$\text{(iii) } p(\,\mathbf{a}, \mathbf{z} \mid \mathbf{b}\,) \neq p(\,\mathbf{a} \mid \mathbf{b}\,)p(\,\mathbf{z} \mid \mathbf{b}\,), \quad \text{(iv) } p(\,\mathbf{b}, \mathbf{z} \mid \mathbf{a}\,) \neq p(\,\mathbf{b} \mid \mathbf{a}\,)p(\,\mathbf{z} \mid \mathbf{a}\,), \tag{1}$$

where in Eq. 1(i) and 1(ii) we interpret the specification of relational property in Definition 1.1—learning relationships irrespective of the absolute properties—as meaning *statistical independence*, while in Eq. 1(iii) and 1(iv) we ensure r.v. $\mathbf{z}$ contains relevant (relationship) information that further informs $\mathbf{a}$ and $\mathbf{b}$ about one another, i.e., $H(\mathbf{a} \mid \mathbf{b}, \mathbf{z}) < H(\mathbf{a} \mid \mathbf{b})$, $H(\mathbf{b} \mid \mathbf{a}, \mathbf{z}) < H(\mathbf{b} \mid \mathbf{a})$ where $H(\cdot \mid \cdot)$ is the conditional entropy. It is easy to see that the following conditions are necessary for r.v. $\mathbf{z}$ to exist: (1) $H(\mathbf{b} \mid \mathbf{a}) > 0$ and $H(\mathbf{a} \mid \mathbf{b}) > 0$, i.e., $\mathbf{a}$ and $\mathbf{b}$ cannot be fully determined by each other; (2) r.v. $\mathbf{a}$, $\mathbf{b}$, $\mathbf{z}$ are not mutually independent, i.e., $p(\,\mathbf{a}, \mathbf{b}, \mathbf{z}\,) \neq p(\,\mathbf{a}\,)p(\,\mathbf{b}\,)p(\,\mathbf{z}\,)$. Our goal for relational learning is to learn about relational property $\mathbf{z}$ that satisfies Eq. 1 in a completely unsupervised fashion. A motivating example for Eq. 1 is provided in Appendix A.1.

In addition, we are interested in two related relational processing tasks: *relational discrimination* and *relational mapping* defined as (VandenBos & APA, 2007):

**Definition 2.1** (**Relational discrimination** in condition). *A discrimination based on the relationship between or among stimuli rather than on absolute features of the stimuli.*

**Definition 2.2** (**Relational mapping**). *The ability to apply what one knows about one set of elements to a different set of elements.*

Relational discrimination allows us to differentiate $(\mathbf{a}^{(i)}, \mathbf{b}^{(i)})$ from $(\mathbf{a}^{(j)}, \mathbf{b}^{(j)})$ based on their relational properties. And relational mapping allows us to apply the relational property of $(\mathbf{a}^{(i)}, \mathbf{b}^{(i)})$ to a different set of data, for example, deduce that $\mathbf{b}^{(j)}$ is related to $\mathbf{a}^{(j)}$ in the same way that $\mathbf{b}^{(i)}$ is related to $\mathbf{a}^{(i)}$.

## 3 METHOD

Learning and inference relational property $\mathbf{z}$ that satisfies all four constraints in Eq. 1 is a challenging problem due to the hard independence constraints in Eq. 1(i) and 1(ii). To overcome this challenge, we first introduce VRL as a tractable learning method that satisfies 3 (out of 4) constraints in Eq. 1—Eq. 1(i), 1(iii), 1(iv). We then discuss VRL's unique optimization challenges, which are partially attributable to its relaxation of the independence requirement in Eq. 1(ii).

### 3.1 VARIATIONAL RELATIONAL LEARNING

The proposed VRL method consists of two parts: first, we encapsulate the relational learning problem with a PGM, called VRL-PGM; we then formulate various relational processing tasks as performing inference and learning in VRL-PGM. The VRL-PGM model, shown in Fig. 1, samples data $\mathbf{a}$, $\mathbf{z}$, and $\mathbf{b}$ from parametric families of distributions—$p_\theta(\,\mathbf{a}\,)$, $p_\theta(\,\mathbf{z}\,)$, $p_\theta(\,\mathbf{b} \mid \mathbf{a}, \mathbf{z}\,)$—that are differentiable almost everywhere with respect to (w.r.t.) $\mathbf{a}$, $\mathbf{z}$, and $\theta$. In practice, we observe only a set of independent realizations $\{\, (\mathbf{a}^{(i)}, \mathbf{b}^{(i)}) \mid i \in [1..N] \,\}$ while the true parameter $\theta^*$ and the corresponding latent variables $\mathbf{z}^{(i)}$ are unobserved. A well-known property of the PGM shown in Fig. 1 is that r.v. $\mathbf{a}$ and $\mathbf{z}$ are *independent* with no variables observed, but *not conditionally independent* when $\mathbf{b}$ is observed, i.e., $p_\theta(\,\mathbf{a}, \mathbf{z}\,) = p_\theta(\,\mathbf{a}\,)p_\theta(\,\mathbf{z}\,)$, $p_\theta(\,\mathbf{a}, \mathbf{z} \mid \mathbf{b}\,) \neq p_\theta(\,\mathbf{a} \mid \mathbf{b}\,)p_\theta(\,\mathbf{z} \mid \mathbf{b}\,)$ (Bishop, 2006). Consequently, VRL-PGM can be viewed as a parametric relational learning model that satisfies 3 (out of 4) constraints in Eq. 1—Eq. 1(i), 1(iii), 1(iv) (note that Eq. 1(iv) is trivially satisfied in VRL-PGM).

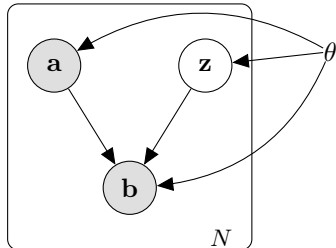

Figure 1: VRL-PGM: a probabilistic graphical model for representing the relational learning problem; the observed r.v. $\mathbf{a}$ and $\mathbf{b}$ are generated from some random process (parameterized by $\theta$) involving a latent r.v. $\mathbf{z}$.

Further discussions on the connection between VRL-PGM and the relational learning problem is provided in Appendix A.2.

Having established VRL-PGM, our primary learning objective is to approximate the unknown true likelihood function $p_\theta(\,\mathbf{b}\mid\mathbf{a},\mathbf{z}\,)$ and posterior $p_\theta(\,\mathbf{z}\mid\mathbf{a},\mathbf{b}\,)$. Learning $p_\theta(\,\mathbf{z}\mid\mathbf{a},\mathbf{b}\,)$ provides us a way to infer $(\mathbf{a}^{(i)},\mathbf{b}^{(i)})$'s relational property $\mathbf{z}^{(i)}$; moreover, it serves as a basis for performing *relational discrimination* where we compare relational properties between different pairs of data. Learning $p_\theta(\,\mathbf{b}\mid\mathbf{a},\mathbf{z}\,)$ allows us to perform *relational mapping* where we use the relational property of $(\mathbf{a}^{(i)},\mathbf{b}^{(i)})$ to map $\mathbf{a}^{(j)}$ to $\mathbf{b}^{(j)}$, i.e., $\mathbf{b}^{(j)}\sim p_\theta(\,\mathbf{b}\mid\mathbf{a}^{(j)},\mathbf{z}^{(i)}\,)$ where $\mathbf{z}^{(i)}\sim p_\theta(\,\mathbf{z}\mid\mathbf{a}^{(i)},\mathbf{b}^{(i)}\,)$.

We estimate the parameter for $p_\theta(\,\mathbf{b}\mid\mathbf{a},\mathbf{z}\,)$ by following the maximum-likelihood (ML) principle, and approximate the true posterior $p_\theta(\,\mathbf{z}\mid\mathbf{a},\mathbf{b}\,)$ with variational Bayesian approach. More specifically, we use a variational distribution $q_\phi(\,\mathbf{z}\mid\mathbf{a},\mathbf{b}\,)$, parameterized by $\phi$, to approximate the unknown (and often intractable) true posterior. Both $\theta$ and $\phi$ are learned through maximizing a variational lower bound, $\mathcal{L}(\theta,\phi;\mathbf{a}^{(i)},\mathbf{b}^{(i)})$ (abbreviated as $\mathcal{L}^{(i)}$), for the conditional log-likelihood $\log p_\theta(\,\mathbf{b}^{(i)}\mid\mathbf{a}^{(i)}\,)$ (derivation is provided in Appendix C):

$$\mathcal{L}^{(i)} = \mathbb{E}_{q_\phi(\mathbf{z}|\mathbf{a}^{(i)},\mathbf{b}^{(i)})}\Big[\log p_\theta(\,\mathbf{b}^{(i)}|\mathbf{a}^{(i)},\mathbf{z}\,) + \log p_\theta(\,\mathbf{z}\,) - \log q_\phi(\,\mathbf{z}|\mathbf{a}^{(i)},\mathbf{b}^{(i)}\,)\Big]. \tag{2}$$

Recall that learning $\mathbf{z}$ independent of $\mathbf{a}$ is central to our relational learning goal. While this independence assumption is built into VRL-PGM, the learning objective $\mathcal{L}^{(i)}$ does not explicitly force $\mathbf{z}$ to be independent of $\mathbf{a}$ nor penalize learning a dependent $\mathbf{z}$. In practice, there may be numerous reasons that could break this independence assumption, e.g., insufficient training data, failure to reach the global optimum, non-identifiability of the model, etc., and it may be desirable to explicitly enforce independence between $\mathbf{z}$ and $\mathbf{a}$. One way to achieve this is to introduce a non-positive function that measures the dependency between $\mathbf{a}$ and $\mathbf{z}$ with maximum attained when they are independent. For example, we can append the negative mutual information between $\mathbf{z}$ and $\mathbf{a}$, $-I(\mathbf{z}\,;\,\mathbf{a}) = -D_{\mathrm{KL}}(p_\theta(\mathbf{z},\mathbf{a})\parallel p_\theta(\,\mathbf{z}\,)p_\theta(\,\mathbf{a}\,))$, to $\mathcal{L}^{(i)}$:

$$\mathcal{L}^{(i)} = \mathbb{E}_{q_\phi(\mathbf{z}|\mathbf{a}^{(i)},\mathbf{b}^{(i)})}\Big[\log p_\theta(\,\mathbf{b}^{(i)}|\mathbf{a}^{(i)},\mathbf{z}\,) + \log p_\theta(\,\mathbf{z}\,) - \log q_\phi(\,\mathbf{z}|\mathbf{a}^{(i)},\mathbf{b}^{(i)}\,)\Big] - I(\mathbf{z}\,;\,\mathbf{a}). \tag{3}$$

Since $I(\mathbf{z}\,;\,\mathbf{a}) \geq 0$ and $I(\mathbf{z}\,;\,\mathbf{a}) = 0$ *if and only if* $\mathbf{z}$ and $\mathbf{a}$ are independent, the addition of $-I(\mathbf{z}\,;\,\mathbf{a})$ to $\mathcal{L}^{(i)}$ not only maintain the validity of the lower bound, but also retain its quality ($\mathbf{z}$ and $\mathbf{a}$ are independent in VRL-PGM).

## 3.2 Optimization challenges

A limitation of the proposed VRL method is its inability to enforce the independence constraint between $\mathbf{z}$ and $\mathbf{b}$ in Eq. 1(ii). This may lead to a worst-case scenario of learning a degenerated posterior $q_\phi(\,\mathbf{z}\mid\mathbf{a},\mathbf{b}\,) = q_\phi(\,\mathbf{z}\mid\mathbf{b}\,)$ where relational property $\mathbf{z}$ only depends on the absolute property of $\mathbf{b}$. Another limitation of VRL is exposed when the assumption $H(\mathbf{b}\mid\mathbf{a}) > 0$ is violated and $\mathbf{b}$ can be completely determined by $\mathbf{a}$. In this case, a degenerated likelihood $p_\theta(\,\mathbf{b}|\mathbf{a},\mathbf{z}\,) = p_\theta(\,\mathbf{b}|\mathbf{a}\,)$ may be fitted and relational property $\mathbf{z}$ plays no role in the learning process. To further explain these limitations and potential mitigation strategies for avoiding the worst-case learning scenario, we first dissect VRL's gradient updating process into the following steps (using a single data point $(\mathbf{a}^{(i)},\mathbf{b}^{(i)})$

as an example): (1) sample $\mathbf{z}^{(i)} \sim q_{\phi_k}(\mathbf{z} \mid \mathbf{a}^{(i)}, \mathbf{b}^{(i)})$ by using the current parameter $\phi_k$; (2) evaluate $\mathcal{L}^{(i)}$ and calculate gradients $g = \nabla_{\theta_k, \phi_k} \mathcal{L}^{(i)}$ by using $\phi_k, \theta_k$; (3) use gradients $g$ to update $\phi_k, \theta_k$ and get new parameters $\phi_{k+1}, \theta_{k+1}$. This process is depicted with an information flow diagram shown in Fig. 2a where ideally we would like every path to contribute to the evaluation of all the terms in its reachable nodes in order to obtain meaningful gradients for updating its associated parameters. However, there are two situations where this is not the case: The first situation, called *information-*

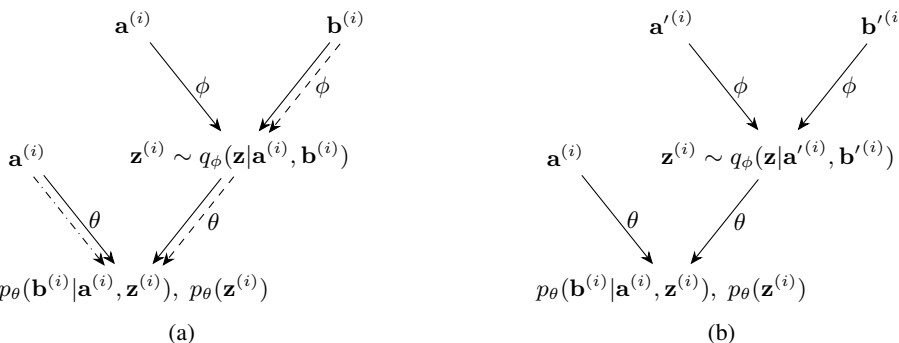

Figure 2: Information flow diagrams depicting VRL's gradients updating process, where each path uses its associated parameters to propagate information in the forward direction and gradients in the backward direction: (a) Overfitting occurs when the learning of $p_\theta(\mathbf{b}^{(i)}|\mathbf{a}^{(i)}, \mathbf{z}^{(i)})$ rely only on the dash-dotted path (*deterministic-mapping*) or the dashed path (*information-shortcut*); (b) Parameter updating process improved with RPDA.

*shortcut*, occurs when the learning of $p_\theta(\mathbf{b}^{(i)} \mid \mathbf{a}^{(i)}, \mathbf{z}^{(i)})$ rely entirely on the dashed path in Fig. 2a; more specifically, the dashed (shortcut) path directly propagates $\mathbf{b}^{(i)}$ through $\mathbf{z}^{(i)}$ to $p_\theta(\mathbf{b}^{(i)}|\mathbf{a}^{(i)}, \mathbf{z}^{(i)})$ and, as a result, the relational property $\mathbf{z}^{(i)}$ may only learn to encode the absolute property of $\mathbf{b}^{(i)}$, i.e., $q_\phi(\mathbf{z} \mid \mathbf{a}^{(i)}, \mathbf{b}^{(i)}) = q_\phi(\mathbf{z} \mid \mathbf{b}^{(i)})$. The second situation, called *deterministic-mapping*, occurs when $\mathbf{b}^{(i)}$ can be fully characterized by $\mathbf{a}^{(i)}$; in this case, the learning of $p_\theta(\mathbf{b}^{(i)} \mid \mathbf{a}^{(i)}, \mathbf{z}^{(i)})$ may only rely on the dash-dotted path in Fig. 2a, i.e., $p_\theta(\mathbf{b}^{(i)}|\mathbf{a}^{(i)}, \mathbf{z}^{(i)}) = p_\theta(\mathbf{b}^{(i)}|\mathbf{a}^{(i)})$. While both situations can be viewed as overfitting problem, *deterministic-mapping* is mainly caused by the data itself and is beyond our control. On the other hand, *information-shortcut* is caused by short-cutting the gradient update process, which we may overcome with additional regularization techniques. Here we propose two approaches for mitigating the *information-shortcut* problem by disrupting the flow of information passing through the shortcut path. In the first approach, we restrict the flow of information by constraining the expressiveness of the latent variable $\mathbf{z}$, e.g., by adopting a discrete categorical r.v. (assuming we know a priori the underlying relational property is discrete). In the second approach, we propose a novel data augmentation strategy—relation-preserving data augmentation (RPDA)—that aims to eliminate the shortcut path entirely. First, we define a set of relation preserving functions $D = \{ d(\mathbf{a}, \mathbf{b}; r) \mid r \in R \text{ (some index set)}, \; d : \mathcal{A} \times \mathcal{B} \to \mathcal{A} \times \mathcal{B} \}$ that preserve data relationship in the following sense: $p_\theta(\mathbf{z} \mid \mathbf{a}, \mathbf{b}) = p_\theta(\mathbf{z} \mid \mathbf{a}', \mathbf{b}'), \; \forall \, (\mathbf{a}', \mathbf{b}') = d(\mathbf{a}, \mathbf{b}; r)$. Assuming we have access to $D$, the proposed RPDA strategy then seek to optimize a modified lower bound $\mathcal{L}_{\text{RPDA}}^{(i)}$:

$$\mathcal{L}_{\text{RPDA}}^{(i)} = \mathbb{E}_{q_\phi(\mathbf{z}|\mathbf{a}'^{(i)}, \mathbf{b}'^{(i)})} \left[ \log p_\theta(\mathbf{b}^{(i)}|\mathbf{a}^{(i)}, \mathbf{z}) + \log p_\theta(\mathbf{z}) - \log q_\phi(\mathbf{z}|\mathbf{a}'^{(i)}, \mathbf{b}'^{(i)}) \right] \quad (4)$$

$$\text{where} \quad (\mathbf{a}'^{(i)}, \mathbf{b}'^{(i)}) = d(\mathbf{a}^{(i)}, \mathbf{b}^{(i)}; r^{(i)}), \; r^{(i)} \sim \mathcal{U}(R).$$

Note that due to the relation preserving property of $D$, we have $q_\phi(\mathbf{z}^{(i)} \mid \mathbf{a}'^{(i)}, \mathbf{b}'^{(i)}) = q_\phi(\mathbf{z}^{(i)} \mid \mathbf{a}^{(i)}, \mathbf{b}^{(i)})$, and thus $\mathcal{L}_{\text{RPDA}}^{(i)}$ is equivalent to $\mathcal{L}^{(i)}$ in Eq. (2). When we optimize with $\mathcal{L}_{\text{RPDA}}^{(i)}$, the gradient update process can be redrawn in Fig. 2b, where now the learning of $p_\theta(\mathbf{b}^{(i)} \mid \mathbf{a}^{(i)}, \mathbf{z}^{(i)})$ can no longer rely solely on the shortcut path to propagate $\mathbf{b}'^{(i)}$ since it differs from $\mathbf{b}^{(i)}$ by a non-deterministic factor $r^{(i)}$. In practice, it may seem unrealistic to assume that we can construct a set of RPDA functions $D$ without extensive knowledge of the underlying relational property. However, we can treat data augmentation as a form of regularization and construct a $D$ that reflects

our prior knowledge and belief of the underlying system (Ronneberger et al., 2015; Perez & Wang, 2017). For example, if we want the learning to be rotation invariance—a common theme in computer vision applications—we can construct a $D$ that consists of image rotation augmentations, e.g., $d(\mathbf{a}, \mathbf{b}; r) = (\text{rot}(\mathbf{a}, r), \text{rot}(\mathbf{b}, r))$ where $\text{rot}(\mathbf{x}, r)$ rotates the image $\mathbf{x}$ by $r \in R = [0, 360)$ degrees (note that both $\mathbf{a}$ and $\mathbf{b}$ are rotated by the same amount). Another example may be, for time-series data of a linear time-invariant (LTI) system (commonly assumed in signal processing and control theory (Oppenheim & Schafer, 2009)), we can construct $D$ with time delay and amplitude scaling, e.g., $d(\mathbf{a}[t], \mathbf{b}[t]; \alpha, \tau) = (\alpha\mathbf{a}[t - \tau], \alpha\mathbf{b}[t - \tau])$, $\alpha, \tau \in R = \mathbb{R} \times \mathbb{Z}$. Additional remarks on the practical applicability of RPDA is provided in Appendix A.3. The full training procedure for VRL with RPDA is summarized in Appendix B.

## 4 RELATED WORK

Existing work on relational learning (Definition 1.1) are mostly based on supervised learning approaches (Lu et al., 2012; Puebla et al., 2021). The rapid growing field of statistical relational learning (SRL) concerns domain models with relational structure (Getoor & Taskar, 2007; De Raedt et al., 2016). Popular SRL methods for link prediction and classification that are based on node attributes similarity (Taskar et al., 2004; Yu et al., 2007) directly contradicts our goal of learning a relational property that is independent of absolute property (node attributes). Other SRL methods based on matrix factorization (Singh & Gordon, 2008; Menon & Elkan, 2011), probabilistic logic programming (De Raedt et al., 2015; Manhaeve et al., 2018), or probabilistic relational models (Friedman et al., 1999; Getoor et al., 2007) typically require some supervision for learning (e.g., labeled examples, known dependency type, etc.) which we do not assume in our work; moreover, most of these methods allow, even encourage, data relationship to depend on its absolute property. Other recent work focus on high-level cognitive tasks, such as visual Q&A and state prediction for complex-physics systems, and derive their relational processing capabilities from learning with clever designed neural networks (Wu et al., 2015; Reed et al., 2015; Fragkiadaki et al., 2016; Chang et al., 2017; Battaglia et al., 2016; Santoro et al., 2017; Battaglia et al., 2018; van Steenkiste et al., 2018; Kipf et al., 2018). Our work differ from these methods in two fundamental ways: (1) we focus on learning an *independent* relational property in a completely *unsupervised* fashion; (2) we derive our relational learning capability from a PGM formulation, which gives us the flexibility to use any compatible inference method or function approximation.

Many existing unsupervised learning methods can also be applied to our problem setting (Song et al., 2007; Mikolov et al., 2013a;b; Kingma & Welling, 2014; Goodfellow et al., 2014; Makhzani et al., 2015; Dilokthanakul et al., 2017); however, most of these methods learn a single representation with superimposing information about the relational and absolute property. The difficulty of decoupling the relational property from the learned representation constitute a major obstacle to relational learning. A closely related method is proposed by Guu et al. (Guu et al., 2018), but their learned latent "edit vector" (loosely related to our relational property) may be coupled with absolute property.

Other related work include methods on learning a disentangled representations with applications in style-transfer, image-to-image translation, domain adaptation, etc. (Bousmalis et al., 2016; Mathieu et al., 2016; Chen et al., 2016; Higgins et al., 2017; Denton & Birodkar, 2017; Huang et al., 2018; Chen et al., 2019). Most of these methods strive to learn a disentangled representations of *content* and *style* (or *pose* for video sequence data) where *content* is generically defined as the underling spatial structure, and *style* as the rendering of the structure. In comparison, our work can be viewed as learning a disentangled representations of *relational* and *absolute property*. We argue that *style-content* separation is fundamentally different from *relational-absolute* separation; more specifically, we consider both *style* and *content* information as *absolute property* (both describe features of an individual data), while *relational property* provides additional information about data relationships.

## 5 EXPERIMENT

In this section, we present experimental results from applying VRL to a variety of relational learning tasks. In our implementation, we parameterize $p_\theta(\mathbf{b} \mid \mathbf{a}^{(i)}, \mathbf{z}^{(i)})$ and $q_\phi(\mathbf{z} \mid \mathbf{a}'^{(i)}, \mathbf{b}'^{(i)})$ with fully-connected networks (MLPs) $f_\theta^p(\mathbf{a}^{(i)}, \mathbf{z}^{(i)})$ and $f_\phi^q(\mathbf{a}'^{(i)}, \mathbf{b}'^{(i)})$, respectively. For binary valued data, we let $p_\theta(\mathbf{b} \mid \mathbf{a}^{(i)}, \mathbf{z}^{(i)})$ be a multivariate Bernoulli distribution whose probability parameters are

computed from $f_\theta^p(\mathbf{a}^{(i)}, \mathbf{z}^{(i)})$. For real-valued data, we let $p_\theta(\mathbf{b}\,|\,\mathbf{a}^{(i)}, \mathbf{z}^{(i)})$ be a multivariate Gaussian distribution with a fixed diagonal covariance and the mean is computed from $f_\theta^p(\mathbf{a}^{(i)}, \mathbf{z}^{(i)})$. We experimented with both continuous and discrete r.v. $\mathbf{z}$. For continuous $\mathbf{z}$, we let the prior $p_\theta(\mathbf{z})$ be a bivariate normal distribution $\mathbf{z} \sim \mathcal{N}(\mathbf{0}, \mathbf{I})$ and $q_\phi(\mathbf{z}\,|\,\mathbf{a}'^{(i)}, \mathbf{b}'^{(i)})$ be a bivariate Gaussian distribution with diagonal covariance whose mean and covariance are computed from $f_\phi^q(\mathbf{a}'^{(i)}, \mathbf{b}'^{(i)})$. For discrete $\mathbf{z}$, we adopted two categorical r.v., $\mathbf{z} = [\mathbf{z}_1, \mathbf{z}_2]$, each having a uniform prior over five categories and let $q_\phi(\mathbf{z}\,|\,\mathbf{a}'^{(i)}, \mathbf{b}'^{(i)})$ represents two categorical r.v. reparameterized with Gumbel-Softmax distributions whose class probabilities are computed from $f_\phi^q(\mathbf{a}'^{(i)}, \mathbf{b}'^{(i)})$ (Jang et al., 2017; Maddison et al., 2017). For RPDA, we used random image rotation as data augmentation functions. Parameters $\theta$ and $\phi$ were jointly trained to maximize $\widetilde{\mathcal{L}}_{\mathrm{RPDA}}^{(i)}$ in Eq. (4) using Adam optimizer (Kingma & Ba, 2015). Full implementation details are provided in Appendix E.1 and source code for reproducible results is available online[1].

## 5.1 RELATIONAL LEARNING WITH DECOUPLED RELATIONSHIPS

We first present two relational learning tasks designed with the MNIST (LeCun & Cortes, 2010) and Omniglot (Lake et al., 2015) datasets (dataset information and data preprocessing is described in Appendix D). A paired MNIST (and Omniglot) dataset $\mathbf{X}^M$ (and $\mathbf{X}^O) = \{\,(\mathbf{a}^{(i)}, \mathbf{b}^{(i)}) \mid i \in [1..N]\,\}$ is constructed where each $\mathbf{a}^{(i)}$ is a randomly rotated MNIST (and Omniglot) binary image and $\mathbf{b}^{(i)}$ is another random counter-clockwise rotation of $\mathbf{a}^{(i)}$ by $0°$, $72°$, $144°$, $216°$, or $288°$. There are five uniquely defined relative rotational relationships between $(\mathbf{a}^{(i)}, \mathbf{b}^{(i)})$ in $\mathbf{X}^M$ (and $\mathbf{X}^O$); furthermore, the relative relationships are decoupled from their absolute properties, i.e., the relationship cannot be inferred from $\mathbf{a}^{(i)}$ or $\mathbf{b}^{(i)}$ alone. The goal of relational learning is to discover the underlying relative rotational relationship. We first demonstrate VRL's *relational discrimination* and *relational mapping* capabilities on the Omniglot relational learning task:

**Relational discrimination.** We used the trained posterior $q_\phi(\mathbf{z}\,|\,\mathbf{a}, \mathbf{b})$ to infer the relational property of a hold-out dataset constructed from the evaluation alphabets. Figure 3a shows a scatter plot of the relational property inferred by VRL where we can see that the approximated posterior accurately cluster (discriminate) data with the same (different) relative rotational relationship together (apart).

**Relational mapping.** We used the trained likelihood $p_\theta(\mathbf{b}\,|\,\mathbf{a}, \mathbf{z})$ to generate images from a given $\mathbf{a}$ and $\mathbf{z}$. We chose $\mathbf{a}$ from a hold-out dataset and $\mathbf{z}$ from: (1) direct sampling in the latent space; (2) relational property inferred from a source data point $(\mathbf{a}_s, \mathbf{b}_s)$. Figure 3b shows predicted images with $\mathbf{z}$ sampled from the latent space shown in Fig. 3a. Figure 3c shows examples of relational mappings from $\mathbf{a}^{(c)}$ to $\mathbf{b}^{(r,c)}$ by applying the relational property inferred from a source data point $(\mathbf{a}_s, \mathbf{b}_s^{(r)})$.

Next, we compare VRL with other unsupervised and self-supervised learning methods: Spatial transformer networks (STN) (Jaderberg et al., 2015) (we tested two variations of STN that uses its localization network to learn about affine transformation, denoted as STN-affine, and rotation transformation, denoted as STN-rotate); Variational autoencoder (VAE) (Kingma & Welling, 2014); VAE with Gaussian mixture prior (GMVAE) (Dilokthanakul et al., 2017); Adversarial autoencoder (AAE) (Makhzani et al., 2015); VAE with added contrastive latent loss (VAE-contrastive) (Chen et al., 2020); self-supervised representation learning method BYOL (Grill et al., 2020); Learning linear structure in representation space through vector arithmetic (Vec-arithmetic) (Mikolov et al., 2013a; Radford et al., 2016); Learning independent causal mechanisms (LICM) (Parascandolo et al., 2018); Neural relational inference (NRI) with graph neural networks (Kipf et al., 2018). VAE, GMVAE, AAE, VAE-contrastive, BYOL, and Vec-arithmetic all trained with dim($\mathbf{z}$)=10. Each baseline method is assessed on both MNIST and Omniglot relational learning tasks through unsupervised clustering. For fair comparison, we adopted image rotation augmentation during the training of all the baseline methods. For VAE-contrastive and BYOL, we used random image rotation augmentations to generate "positive paris" for training. In our comparison, STN-affine and STN-rotate are considered as the benchmark methods as they are architecturally designed to learn about the spatial transformation between $\mathbf{a}^{(i)}$ and $\mathbf{b}^{(i)}$. In contrast, VRL does not make any assumption about the underlying relationship. Implementation details for the baseline methods are provided in Appendix E.2

---

[1]https://github.com/kh1iu/vrl.git

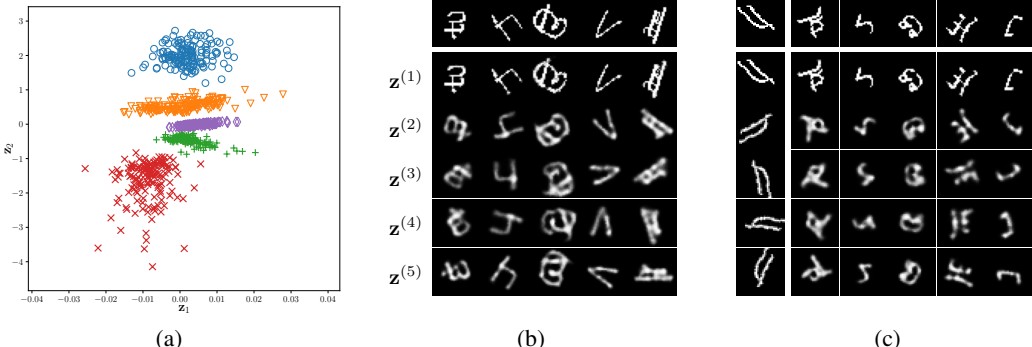

(a)          (b)          (c)

Figure 3: Learning decoupled relative rotational relationship with VRL: (a) Scatter plots of 2-D relational property of $\mathbf{X}^O$ inferred by VRL (relative rotational relationship labels: $\bigcirc : 0°$, $\nabla : 72°$, $+ : 144°$, $\times : 216°$, $\diamondsuit : 288°$); (b) Images predicted from sampled latent variables (sampling the centroid of each cluster in (a): "$\bigcirc$" $\rightarrow \mathbf{z}^{(1)}$, "$\nabla$" $\rightarrow \mathbf{z}^{(2)}$, "$+$" $\rightarrow \mathbf{z}^{(3)}$, "$\times$" $\rightarrow \mathbf{z}^{(4)}$, "$\diamondsuit$" $\rightarrow \mathbf{z}^{(5)}$); each image $\mathbf{b}^{(r,c)}, 1 \le r, c \le 5$, was predicted from $\mathbf{a}^{(c)}$ (shown in the top row) and $\mathbf{z}^{(r)}$ by $\mathbf{b}^{(r,c)} \sim p_\theta(\mathbf{b} \mid \mathbf{a}^{(c)}, \mathbf{z}^{(r)})$; (c) Relational mappings of top row images by applying relational property inferred from pairs of source images $(\mathbf{a}_s, \mathbf{b}_s^{(r)})$ shown in the left-most column with $\mathbf{a}_s, \mathbf{b}_s^{(1)}, ..., \mathbf{b}_s^{(5)}$ arranged from top to bottom; each image $\mathbf{b}^{(r,c)}, 1 \le r, c \le 5$ was generated by $\mathbf{b}^{(r,c)} \sim p_\theta(\mathbf{b} \mid \mathbf{a}^{(c)}, \mathbf{z}^{(r)})$ where $\mathbf{z}^{(r)} \sim q_\phi(\mathbf{z} \mid \mathbf{a}_s, \mathbf{b}_s^{(r)})$.

To quantitatively evaluate the clustering performance, we calculate the classification error rate of a hold-out dataset based on a simple classifier that was trained on 5% of the data with label. For continuous latent space, we trained a standard multi-class support vector machine classifier with radial basis function kernel (Manning et al., 2008). For discrete (categorical) latent space, we follow the evaluation protocol on clustering assignment described in Makhzani et al. (2015). Results are summarized in Table 1 where each entry reports the average and standard deviation over 5 runs with different random seeds and each run consists of either 3 (randomly selected out of 5) or 5 relative rotational relationships. We can see that the proposed VRL with RPDA achieved high accuracy in recovering the relative relationships for all the tasks and outperformed other baseline methods including the benchmark STN-affine and STN-rotate. To gain insight into the *information-shortcut* problem and the proposed mitigation strategies introduced in Sec. 3.2, we performed an ablation study on the prior selection of $\mathbf{z}$ and RPDA. The results are summarized in Table 1 and we make the following observations: First, representing $\mathbf{z}$ as a discrete categorical (Cat.) r.v. limits its expressiveness and improves VRL's performance over a continuous prior (Cont.); Second, RPDA is a critical component necessary for VRL to learn a meaningful and independent relational property, especially when flexible function approximations such as deep neural networks are used. Additional experimental results and analysis are provided in Appendix F.

## 5.2 RELATIONAL LEARNING WITH COUPLED RELATIONSHIPS

To further test the robustness and generalizability of VRL, we consider a scenario where the relative relationship is coupled with the absolute property. To setup this experiment, we modified the construction of $\mathbf{X}^M$, denoted as $\mathbf{X}^{M_d}$, so that the relative rotational relationship between $\mathbf{a}^{(i)}$ and $\mathbf{b}^{(i)}$ is completely determined by its digit representation (absolute property of $\mathbf{a}^{(i)}$ and $\mathbf{b}^{(i)}$): $\mathbf{a}^{(i)} \in [\text{'0', '1'}] \rightarrow 0°, \mathbf{a}^{(i)} \in [\text{'2', '3'}] \rightarrow 72°, \mathbf{a}^{(i)} \in [\text{'4', '5'}] \rightarrow 144°, \mathbf{a}^{(i)} \in [\text{'6', '7'}] \rightarrow 216°, \mathbf{a}^{(i)} \in [\text{'8', '9'}] \rightarrow 288°$ (read: if $\mathbf{a}^{(i)}$ is digit '0' or '1' then the relative rotational relationship between $(\mathbf{a}^{(i)}, \mathbf{b}^{(i)})$ is $0°$). The question then arise: is VRL capable of learning an independent relational property even when the underlying relative relationship is coupled with the absolute property? To test this idea, we trained VRL on $\mathbf{X}^{M_d}$ but validated it on a hold-out dataset of $\mathbf{X}^M$ which has a decoupled relative relationship. The results are shown in Fig. 4, where we can see that VRL was indeed capable of learning an independent relational property irrespective of the digit representation. If this were not the case, we would expect to see a scatter plot Fig. 4a with heavily overlapped

Table 1: Unsupervised clustering accuracy (in %) for MNIST and Omniglot relational learning tasks.

| | MNIST | | Omniglot | |
|---|---|---|---|---|
| Num. relationships | 3 | 5 | 3 | 5 |
| VRL (Cat., no RPDA) | **93.9**±6.7 | 46.9±7.4 | **99.9**±0.1 | 22.4±0.7 |
| VRL (Cat., RPDA) | **99.8**±0.1 | **99.9**±0.1 | **99.6**±0.3 | **99.9**±0.1 |
| VRL (Cont., no RPDA) | 60.1±7.1 | 28.3±4.2 | **96.4**±6.7 | 32.1±4.9 |
| VRL (Cont., RPDA) | **99.8**±0.2 | **99.9**±0.1 | **99.8**±0.1 | **97.7**±0.6 |
| STN-affine | 33.7±1.1 | 19.8±0.9 | 34.1±0.4 | 19.9±0.8 |
| STN-rotate | **98.3**±1.3 | **97.2**±0.8 | 47.4±26.1 | 30.2±13.1 |
| VAE | **93.8**±3.5 | 60.6±2.3 | **98.2**±1.9 | 63.0±3.7 |
| GMVAE | 73.9±3.7 | 57.2±2.5 | 47.7±1.4 | 38.8±1.6 |
| AAE | 49.1±1.2 | 31.1±2.4 | 66.5±4.2 | 33.3±1.2 |
| VAE-contrastive | **92.4**±3.8 | 60.5±2.7 | **93.4**±6.3 | 59.4±4.4 |
| BYOL | 49.1±12.7 | 27.7±2.4 | 36.7±1.7 | 23.3±1.2 |
| Vec-arithmetic | 64.7±11.9 | 54.3±1.6 | 53.5±14.1 | 50.8±1.3 |
| LICM | 36.6±3.3 | 19.7±1.1 | 35.0±2.5 | 21.6±2.3 |
| NRI | 34.5±1.2 | 20.5±1.0 | 34.6±0.9 | 21.1±0.9 |

relative relationship labels since $\mathbf{X}^M$ was constructed with random relative rotational relationships. Figures 4b and 4c further demonstrate VRL generalizes well to unseen data—VRL trained on $\mathbf{X}^{M_d}$ learned to rotate *any* digit by *any* amount despite not having seen most of the digit-rotation pairs during training.

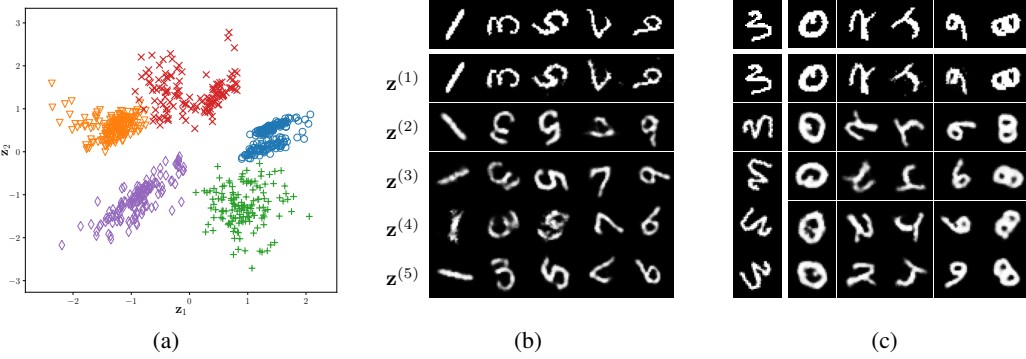

| (a) | (b) | (c) |

Figure 4: Learning coupled relative rotational relationship with VRL: (a) Scatter plot of 2-D relational property of $\mathbf{X}^M$ inferred by VRL (relative rotational relationship labels: $\bigcirc : 0°$, $\triangledown : 72°$, $+ : 144°$, $\times : 216°$, $\diamond : 288°$); (b) Images predicted from sampled latent variables (sampling the centroid of each cluster in (a): "$\bigcirc$" $\to \mathbf{z}^{(1)}$, "$\triangledown$" $\to \mathbf{z}^{(2)}$, "$+$" $\to \mathbf{z}^{(3)}$, "$\times$" $\to \mathbf{z}^{(4)}$, "$\diamond$" $\to \mathbf{z}^{(5)}$); (c) Relational mappings of top row images by applying relational property inferred from pairs of source images $(\mathbf{a}_s, \mathbf{b}_s^{(r)})$ in the left-most column.

## 5.3 RELATIONAL LEARNING WITH HIGH-LEVEL PERCEPTION REASONING

Finally, we present results of using VRL to learn about relative facial expression changes, facial illumination condition changes, and speech emotion changes. We extracted images of three facial expressions (happy, surprised, sad) of each subject from the Yale Face Database (Belhumeur et al., 1997) to form a paired dataset $\mathbf{X}^{F_e}$ where each data point $(\mathbf{a}^{(i)}, \mathbf{b}^{(i)})$ represents a subject with different facial expressions. Next, we extracted images of four illumination conditions (left (L), front (F), right (R), top (T)) of each subject from the Extended Yale Face Database B (Georghiades et al., 2001) to form a paired dataset $\mathbf{X}^{F_l}$ where each data point $(\mathbf{a}^{(i)}, \mathbf{b}^{(i)})$ represents a subject with

different illumination conditions. Lastly, we extracted speech waveforms (represented as log-mel spectrogram) of three emotions (calm, angry, fearful) of each voice actor from the Ryerson Audio-Visual Database of Emotional Speech and Song (RAVDESS) (Livingstone & Russo, 2018) to form a paired dataset $\mathbf{X}^S$ where each data point $(\mathbf{a}^{(i)}, \mathbf{b}^{(i)})$ represents a voice actor with different emotions. Dataset information and data preprocessing is described in Appendix D. Due to the extremely limited data samples in $\mathbf{X}^{F_e}$ (45 unique images), $\mathbf{X}^{F_l}$ (112 unique images), and $\mathbf{X}^S$ (144 unique speech recordings), we focus on learning *undirected* relative relationship, i.e., $(\mathbf{a}^{(i)}, \mathbf{b}^{(i)})$ and $(\mathbf{b}^{(i)}, \mathbf{a}^{(i)})$ have the same relative relationship. In this case, $\mathbf{X}^{F_e}$, $\mathbf{X}^{F_l}$, and $\mathbf{X}^S$ consist of 3, 6, and 3 *undirected* relative relationships, respectively. For RPDA, we adopted random image rotation for the VRL training on $\mathbf{X}^{F_e}$ and $\mathbf{X}^{F_l}$, and random time delay and amplitude scaling for $\mathbf{X}^S$ due to its temporal data characteristic; furthermore, since we are only interested in learning *undirected* relationships, we augment RPDA with random swapping operation (see Appendix E.1 for implementation details). Due to the limited datasets, both training and validation are performed on the entire dataset. The inference results from a trained posterior $q_\phi(\mathbf{z} \mid \mathbf{a}, \mathbf{b})$ are shown in Fig. 5 where we can see that VRL is effective in learning a relative relationship that is independent of subject's identity, facial expression, illumination condition, speech content, and emotion. Comparing our results with existing unsupervised methods on facial images (Song et al., 2007; Wu et al., 2013; Shi et al., 2018; Tapaswi et al., 2019) and speech emotion recognition (Neumann & Vu, 2019; Li et al., 2021), we remark that existing methods cluster data by their absolute property (e.g., subject identity, speech emotion, etc.), while VRL clusters data by their relational property (e.g., relative facial expression, illumination, or speech emotion changes). Additional relational mapping results are provided in Appendix F.2.

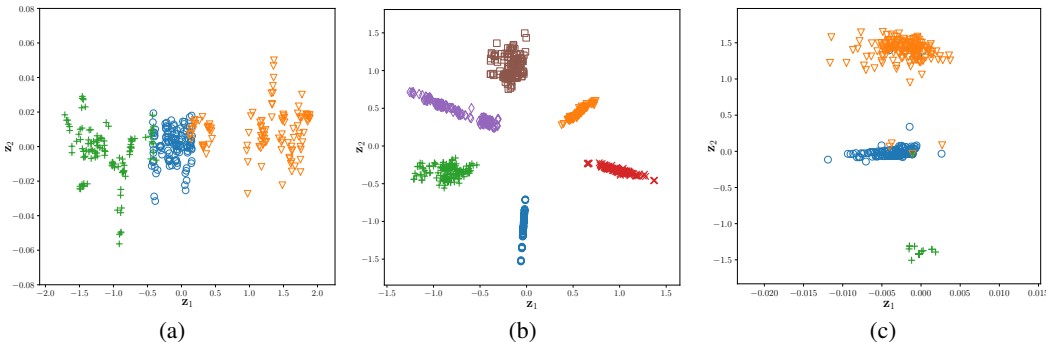

(a)  (b)  (c)

Figure 5: Learning relative relationships between facial images and speech waveforms: (a) Learning relative facial expression changes with 2-D VRL relational property (labels: $\bigtriangledown$:"happy-sad", $\bigcirc$:"happy-surprised", $+$:"surprised-sad"); (b) Learning relative facial illumination condition changes with 2-D VRL relational property (labels: $\bigtriangledown$:"L-R, $\diamondsuit$:"F-T, $\bigcirc$: "L-F, $+$:"L-T, $\times$:"F-R, $\square$:"R-T); (c) Learning relative speech emotion changes with 2-D VRL relational property (labels: $\bigtriangledown$:"calm-fearful", $\bigcirc$:"calm-angry", $+$:"angry-fearful")

## 6 DISCUSSION AND CONCLUSION

The proposed VRL method comes with both advantages and disadvantages: the main advantage of VRL is its relational learning capabilities; however, this may also be one of its disadvantages. More specifically, VRL can learn an independent relational property even when it is coupled with the absolute property (see Sec. 5.2), i.e., VRL is oblivious to the coupling information between the two properties. Nevertheless, such information may be of interest to the user, and in this regard, VRL only provides a partial view of the data.

In conclusion, the proposed VRL method is an efficient and effective unsupervised learning method for addressing the relational learning problem where our goal is to learn an independent relational property. By dissecting the data information into decoupled *relational* and *absolute property*, we hope VRL can bring new insight into everyday data analysis and ultimately find applications for a wide variety of problems.

ACKNOWLEDGMENT

We thank anonymous reviewers for comments that helped improve the paper. We would also like to thank Huseyin Denli, Antonio Paiva, Ashutosh Tewari, Myun-Seok Cheon, and Peng Xu (ExxonMobil) for insightful discussion and valuable feedback.

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

# A  ADDITIONAL REMARKS

## A.1  REMARKS ON EQUATION 1

To provide additional insight into Eq. (1), we give the following example (attributed to S. Bernstein (Hogg et al., 2005)): Let $\mathbf{a}$ and $\mathbf{b}$ be two independent tosses of a fair coin, where we designate 1 for heads and 0 for tails. Let a third random variable $\mathbf{z}$ be equal to 1 if exactly one of two coin tosses resulted in "heads", and 0 otherwise. It is easy to see that this example r.v. $\mathbf{a}$, $\mathbf{b}$ ans $\mathbf{z}$ satisfy all four conditions in Eq. (1) and we can intuitively interpret $\mathbf{z}$ as meaning "different tosses" which is a relational property since it indicates the relationship between $\mathbf{a}$ and $\mathbf{b}$ (same vs. different) and is independent of $\mathbf{a}$ and $\mathbf{b}$, i.e., $\mathbf{z}$ is undertermined when observing $\mathbf{a}$ or $\mathbf{b}$ alone. The main difference between this example and our definition in Eq. (1) is that we do not specify what $\mathbf{z}$ is; we only assume the existence of $\mathbf{z}$ and our goal is to learn about it.

## A.2  REMARKS ON VRL-PGM

The proposed VRL-PGM reflects our priority and compromise for using a PGM to represent the relational learning problem: we sacrifice some identifiability of the original abstract problem but obtain a rigorous and tractable model that achieves our objective of learning an independent relational property. Here we discuss aspects of the original relational learning problem (see Section 2) that differ from the proposed VRL-PGM (see Section 3.1). First, the original problem description in Eq. 1 specifies that the relational property $\mathbf{z}$ be independent of *both* $\mathbf{a}$'s and $\mathbf{b}$'s absolute properties; however, learning and inferencing relational property $\mathbf{z}$ that satisfies both independence constraints is a challenging problem; therefore, as a compromise, VRL-PGM only enforce the independence constraint between $\mathbf{z}$ and $\mathbf{a}$. Second, the original problem is inherently undirected with no cause-effect relationship between $\mathbf{a}$ and $\mathbf{b}$, whereas VRL-PGM is based a directed acyclic graph (DAG) that artificially introduces conditional dependency between $\mathbf{a}$ and $\mathbf{b}$. We note that this can be viewed as the result of VRL-PGM choosing to satisfy Eq. 1(i), 1(iii), 1(iv). It is equally valid for VRL-PGM to choose to satisfy Eq. 1(ii), 1(iii), 1(iv) and it would lead to the same model but with $\mathbf{a}$ and $\mathbf{b}$ swapped. In other words, the application of VRL does not require the true conditional dependency between $(\mathbf{a}, \mathbf{b})$ be known in advance only that it is maintained consistently throughout learning and inference. The above-mentioned discrepancies represent the compromises we made with adopting VRL-PGM in exchange for a rigorous and tractable method for learning an independent relational property. Finally, these compromises are not been made without consequences: *information-shortcut* is a direct result of not enforcing independence constraint between $\mathbf{z}$ and $\mathbf{b}$, and *deterministic-mapping* can be viewed as caused by the causal relationship VRL-PGM introduced between $\mathbf{a}$ and $\mathbf{b}$.

## A.3  REMARKS ON RPDA

Here we discuss the practical applicability of the proposed RPDA strategy. More specifically, we argue that in many practical problem settings, the RPDA functions $D$ can be designed without any knowledge of the underlying relational property. For example, as we have explained in Sec. 3.2, in many computer vision applications, rotation invariant is a desirable property for the learned model; for example, in spectral imaging applications, oftentimes the orientation of the images are not preserved or not enforced only that they are consistent between the same paired images (Ronneberger et al., 2015). In such problem setting, we can safely use image rotation function for constructing $D$. Another example may be: for a discrete time-series data $\mathbf{a}[t]$, $\mathbf{b}[t]$ that represent the input and output of a linear time-invariant (LTI) system (commonly assumed in signal processing and control theory (Oppenheim & Schafer, 2009)), and we want to learn a relational property that characterize the system's impulse response. We have $\alpha \mathbf{b}[t - \tau] = \alpha \mathbf{a}[t - \tau]$, $\forall \alpha \in \mathbb{R}, \tau \in \mathbb{Z}$, and we can construct $D$ with $d(\mathbf{a}[t], \mathbf{b}[t]; \alpha, \tau) = (\alpha \mathbf{a}[t - \tau], \alpha \mathbf{b}[t - \tau]), \alpha, \tau \in R = \mathbb{R} \times \mathbb{Z}$. In all of the above examples, the construction of RPDA functions $D$ reflects our prior knowledge and belief of the underlying system and not based on the underlying relational property. However, just like any data augmentation, the effectiveness of RPDA will depend on the problem setting and we advocate to start without RPDA and only apply it when suspecting information-shortcut occurs.

# B VRL ALGORITHM

---
**Algorithm 1** VRL with RPDA
---

   **procedure** VRL($\mathbf{X}$, $p(\boldsymbol{\epsilon})$, $D$)        ▷ If RPDA not available, $D = \{\, \mathrm{id}(\cdot) \mid (\mathbf{a}, \mathbf{b}) = \mathrm{id}(\mathbf{a}, \mathbf{b}) \,\}$
       Initialize parameters $\theta, \phi$
       **while** not convergence of parameters $(\theta, \phi)$ **do**
          Sample minibatch $\{\, (\mathbf{a}^{(i)}, \mathbf{b}^{(i)}) \mid i \in [1..M] \,\}$ from $\mathbf{X}$.
          Run RPDA and obtain $(\mathbf{a}'^{(i)}, \mathbf{b}'^{(i)}) = d(\mathbf{a}^{(i)}, \mathbf{b}^{(i)}; r^{(i)})$, $r^{(i)} \sim \mathcal{U}(R)$, $i = 1, ..., M$.
          Compute gradients $g = \nabla_{\theta, \phi} \mathcal{L}_{\mathrm{RPDA}}^{(i)}$ (see Eq. (4)).
          Update parameters $\theta, \phi$ using gradients $g$ (e.g., SGD).
       **end while**
       **return** $\theta, \phi$
   **end procedure**

---

# C DERIVATION OF VARIATIONAL LOWER BOUND

To derive a variational lower bound for VRL-PGM, we first write the log-evidence as $\log p_\theta(\mathbf{X}) = \log p_\theta(\{\, (\mathbf{a}^{(i)}, \mathbf{b}^{(i)}) \mid i \in [1..N] \,\}) = \sum_{i=1}^{N} \log p_\theta(\mathbf{a}^{(i)}, \mathbf{b}^{(i)})$, where each term in the summation can be expressed as:

$$\log p_\theta(\mathbf{a}^{(i)}, \mathbf{b}^{(i)}) = D_{\mathrm{KL}}\Big( q_\phi(\mathbf{z} \mid \mathbf{a}^{(i)}, \mathbf{b}^{(i)}) \,\Big\|\, p_\theta(\mathbf{z} \mid \mathbf{a}^{(i)}, \mathbf{b}^{(i)}) \Big)$$
$$+ \mathbb{E}_{q_\phi(\mathbf{z} \mid \mathbf{a}^{(i)}, \mathbf{b}^{(i)})} \Big[ \log p_\theta(\mathbf{z}, \mathbf{a}^{(i)}, \mathbf{b}^{(i)}) - \log q_\phi(\mathbf{z} \mid \mathbf{a}^{(i)}, \mathbf{b}^{(i)}) \Big]. \quad (5)$$

The first term on the right-hand side (RHS) is the KL-divergence from $p_\theta(\mathbf{z} \mid \mathbf{a}^{(i)}, \mathbf{b}^{(i)})$ to $q_\phi(\mathbf{z} \mid \mathbf{a}^{(i)}, \mathbf{b}^{(i)})$, which provides a measure of dissimilarity between the two distributions; the second term on the RHS continues as:

$$\mathbb{E}_{q_\phi(\mathbf{z} \mid \mathbf{a}^{(i)}, \mathbf{b}^{(i)})} \Big[ \log p_\theta(\mathbf{z}, \mathbf{a}^{(i)}, \mathbf{b}^{(i)}) - \log q_\phi(\mathbf{z} \mid \mathbf{a}^{(i)}, \mathbf{b}^{(i)}) \Big]$$
$$= \mathbb{E}_{q_\phi(\mathbf{z} \mid \mathbf{a}^{(i)}, \mathbf{b}^{(i)})} \Big[ \log p_\theta(\mathbf{b}^{(i)} \mid \mathbf{a}^{(i)}, \mathbf{z}) p_\theta(\mathbf{z}) p_\theta(\mathbf{a}^{(i)}) - \log q_\phi(\mathbf{z} \mid \mathbf{a}^{(i)}, \mathbf{b}^{(i)}) \Big]$$
$$= \mathbb{E}_{q_\phi(\mathbf{z} \mid \mathbf{a}^{(i)}, \mathbf{b}^{(i)})} \Big[ \log p_\theta(\mathbf{b}^{(i)} \mid \mathbf{a}^{(i)}, \mathbf{z}) + \log p_\theta(\mathbf{z}) - \log q_\phi(\mathbf{z} \mid \mathbf{a}^{(i)}, \mathbf{b}^{(i)}) \Big] + \log p_\theta(\mathbf{a}^{(i)}), \quad (6)$$

where in the second line we use the fact that r.v. $\mathbf{a}$ and $\mathbf{z}$ are independent. Substitute Eq. (6) back in (5) and rearrange terms gives us the expression for Eq. 2:

$$\log p_\theta(\mathbf{b}^{(i)} \mid \mathbf{a}^{(i)}) = D_{\mathrm{KL}}\Big( q_\phi(\mathbf{z} \mid \mathbf{a}^{(i)}, \mathbf{b}^{(i)}) \,\Big\|\, p_\theta(\mathbf{z} \mid \mathbf{a}^{(i)}, \mathbf{b}^{(i)}) \Big) + \mathcal{L}^{(i)} \quad (7)$$

where

$$\mathcal{L}^{(i)} = \mathbb{E}_{q_\phi(\mathbf{z} \mid \mathbf{a}^{(i)}, \mathbf{b}^{(i)})} \Big[ \log p_\theta(\mathbf{b}^{(i)} \mid \mathbf{a}^{(i)}, \mathbf{z}) + \log p_\theta(\mathbf{z}) - \log q_\phi(\mathbf{z} \mid \mathbf{a}^{(i)}, \mathbf{b}^{(i)}) \Big].$$

The term $\mathcal{L}^{(i)}$ serves as a lower bound for the conditional log-likelihood $\log p_\theta(\mathbf{b}^{(i)} \mid \mathbf{a}^{(i)})$ since KL-divergence is non-negative. Maximizing $\mathcal{L}^{(i)}$ w.r.t. $\phi$ and $\theta$ gives us both a ML estimate for $p_\theta(\mathbf{b} \mid \mathbf{a}, \mathbf{z})$ (by maximizing the first term inside the expectation in $\mathcal{L}^{(i)}$) and a lower KL-divergence (the better $q_\phi(\mathbf{z} \mid \mathbf{a}^{(i)}, \mathbf{b}^{(i)})$ approximates the true posterior $p_\theta(\mathbf{z} \mid \mathbf{a}^{(i)}, \mathbf{b}^{(i)})$) as the conditional log-likelihood $\log p_\theta(\mathbf{b}^{(i)} \mid \mathbf{a}^{(i)})$ does not depend on $\phi$. The lower bound $\mathcal{L}^{(i)}$ can be maximized with gradient ascend methods; however, its gradients w.r.t. $\phi$ is difficult to obtain: the expectation in $\mathcal{L}^{(i)}$ is taken w.r.t. the distribution $q_\phi(\mathbf{z} \mid \mathbf{a}^{(i)}, \mathbf{b}^{(i)})$, which is a function of $\phi$ (Paisley et al., 2012). To obtain efficient estimators for both $\mathcal{L}^{(i)}$ and its gradients, we adopt the *reparameterization trick* developed in Kingma & Welling (2014) where the r.v. $\mathbf{z}$ is expressed as a transformation of another r.v. $\boldsymbol{\epsilon} \sim p(\boldsymbol{\epsilon})$ that is independent of $\mathbf{a}$, $\mathbf{b}$, and $\phi$: $\mathbf{z} = g(\boldsymbol{\epsilon}, \mathbf{a}^{(i)}, \mathbf{b}^{(i)}, \phi)$ where $g$ is some differentiable and invertible transformation. Given such a change of variable, the lower bound $\mathcal{L}^{(i)}$ can be rewritten as:

$$\mathcal{L}^{(i)} = \mathbb{E}_{p(\boldsymbol{\epsilon})} \Big[ \log p_\theta(\mathbf{b}^{(i)} \mid \mathbf{a}^{(i)}, \mathbf{z}) + \log p_\theta(\mathbf{z}) - \log q_\phi(\mathbf{z} \mid \mathbf{a}^{(i)}, \mathbf{b}^{(i)}) \Big], \quad (8)$$

where $\mathbf{z} = g(\boldsymbol{\epsilon}, \mathbf{a}^{(i)}, \mathbf{b}^{(i)}, \phi)$ and $\boldsymbol{\epsilon} \sim p(\boldsymbol{\epsilon})$. Note that the expectation in Eq. (8) is taken w.r.t. $p(\boldsymbol{\epsilon})$ and we can approximate $\mathcal{L}^{(i)}$ with a Monte Carlo estimator:

$$\widetilde{\mathcal{L}}^{(i)} = \frac{1}{L} \sum_{l=1}^{L} \log p_\theta(\mathbf{b}^{(i)} \mid \mathbf{a}^{(i)}, \mathbf{z}^{(i,l)}) + \log p_\theta(\mathbf{z}^{(i,l)}) - \log q_\phi(\mathbf{z}^{(i,l)} \mid \mathbf{a}^{(i)}, \mathbf{b}^{(i)}), \quad (9)$$

where $\mathbf{z}^{(i,l)} = g(\boldsymbol{\epsilon}^{(i,l)}, \mathbf{a}^{(i)}, \mathbf{b}^{(i)}, \phi)$ and $\boldsymbol{\epsilon}^{(i,l)} \sim p(\boldsymbol{\epsilon})$. An estimator for $\mathcal{L}_{\text{RPDA}}^{(i)}$ in Eq. (4) follows similarly as:

$$\widetilde{\mathcal{L}}_{\text{RPDA}}^{(i)} = \frac{1}{L} \sum_{l=1}^{L} \log p_\theta(\mathbf{b}^{(i)} \mid \mathbf{a}^{(i)}, \mathbf{z}^{(i,l)}) + \log p_\theta(\mathbf{z}^{(i,l)}) - \log q_\phi(\mathbf{z}^{(i,l)} \mid \mathbf{a}'^{(i)}, \mathbf{b}'^{(i)}), \quad (10)$$

$$\text{where} \quad (\mathbf{a}'^{(i)}, \mathbf{b}'^{(i)}) = d(\mathbf{a}^{(i)}, \mathbf{b}^{(i)}; r^{(i)}), \ \ r^{(i)} \sim \mathcal{U}(R),$$

and $\mathbf{z}^{(i,l)} = g(\boldsymbol{\epsilon}^{(i,l)}, \mathbf{a}'^{(i)}, \mathbf{b}'^{(i)}, \phi)$, $\boldsymbol{\epsilon}^{(i,l)} \sim p(\boldsymbol{\epsilon})$. The lower bound for a minibatches of data $\mathbf{X}^M = \{(\mathbf{a}^{(i)}, \mathbf{b}^{(i)}) \mid i \in [1..M]\}$ can be approximated by $\widetilde{\mathcal{L}}_{\text{RPDA}}(\theta, \phi; \mathbf{X}^M) = \frac{N}{M} \sum_{i=1}^{M} \widetilde{\mathcal{L}}_{\text{RPDA}}^{(i)}$. And finally, the gradients $\nabla_{\theta,\phi} \widetilde{\mathcal{L}}_{\text{RPDA}}(\theta, \phi; \mathbf{X}^M) = \frac{N}{M} \sum_{i=1}^{M} \nabla_{\theta,\phi} \widetilde{\mathcal{L}}_{\text{RPDA}}^{(i)}$ can be computed in a straightforward manner and used to update the parameters $\theta$ and $\phi$ with stochastic optimization methods, such as SGD.

## D  DATASET AND DATA PREPROCESSING

We used the following datasets in our experiments:

**MNIST** dataset (available under CC BY-SA 3.0 license) consists of $28 \times 28$ grayscale images of standard handwritten digits with labels; the dataset is composed of 60,000 training samples and 10,000 testing samples (LeCun & Cortes, 2010).

**Omniglot** dataset (available under MIT license) contains 1623 different handwritten characters ($105 \times 105$ grayscale images) from 50 different alphabets; the dataset is split into a background (training) set of 30 alphabets and an evaluation (testing) set of 20 alphabets (Lake et al., 2015).

**Yale Face Database** (see (Georghiades, 2001) for dataset permission) contains 165 grayscale face images of 15 human subjects under 8 facial expressions and 3 illumination conditions (Belhumeur et al., 1997).

**Extended Yale Face Database B** (see (Georghiades, 2001) for dataset permission) contains 16128 grayscale face images of 28 human subjects under 9 poses and 64 illumination conditions (Georghiades et al., 2001).

**Ryerson Audio-Visual Database of Emotional Speech and Song (RAVDESS)** (available under CC BY-NC-SA 4.0 license) is a speech and song dataset containing 24 professional actors (12 female, 12 male) vocalizing two lexically-matched statements in a neutral North American accent with wide range of emotions (Livingstone & Russo, 2018).

Each handwritten character image in MNIST and Omniglot dataset is resized to $32 \times 32$ binary image. Each facial image in Yale and Extended Yale dataset is center-cropped, resized to $64 \times 64$, and normalized pixel values to be within $[0, 5]$. Each speech waveform in RAVDESS dataset is trimmed to remove silences both at the start and at the end, and then clipped or zero-padded to be 3 seconds long. We then represent each waveform by its log-amplitude mel spectrogram with 128 mel bands and frame size of 512 samples. Finally, each log-mel spectrogram image is resized to $64 \times 64$. Examples of facial images from Yale and Extended Yale, and log-mel spectrograms of RAVDESS speech waveforms are shown in Fig. 6.

## E  IMPLEMENTATION DETAILS

### E.1  VRL IMPLEMENTATION

Here we describe the details of our VRL implementation. We parameterize both $p_\theta(\mathbf{b} \mid \mathbf{a}^{(i)}, \mathbf{z}^{(i,l)})$ and $q_\phi(\mathbf{z} \mid \mathbf{a}'^{(i)}, \mathbf{b}'^{(i)})$ using fully-connected networks (MLPs) with rectified linear non-linearities.

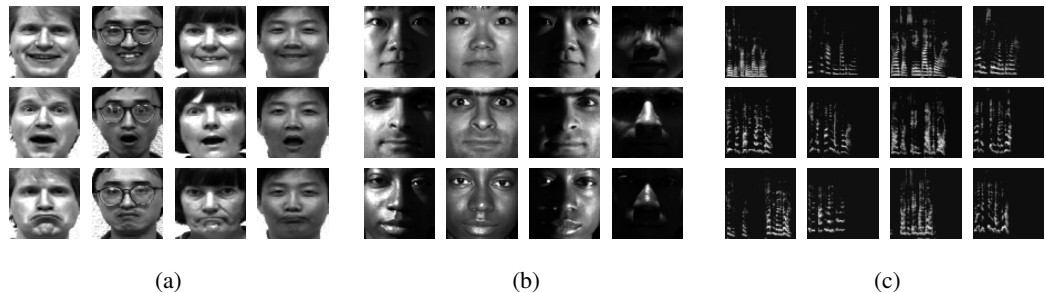

(a)             (b)             (c)

Figure 6: Examples of facial images and speech waveforms: (a) Subjects (columns) with different facial expressions (rows): happy, surprised, sad; (b) Subjects (rows) with different illumination conditions (columns): left, front, right, top; (c) Log-mel spectrogram of actors vocalizing two statements with different speech emotions (Columns: 1. male actor vocalizing "Kids are talking by the door", 2. female actor vocalizing "Kids are talking by the door", 3. male actor vocalizing "Dogs are sitting by the door", 4. female actor vocalizing "Dogs are sitting by the door"; Rows: calm, angry, fearful).

For binary valued data such as MNIST and Omniglot, we let $p_\theta(\mathbf{b} \mid \mathbf{a}^{(i)}, \mathbf{z}^{(i,l)})$ be a multivariate Bernoulli distribution whose probability parameters $\mathbf{p}^{(i,l)}$ are computed from MLPs $f_\theta^p(\mathbf{a}^{(i)}, \mathbf{z}^{(i,l)})$:

$$f_\theta^p : [\mathbf{a}^{(i)}, \mathbf{z}^{(i,l)}] \to \mathrm{FC}(512) \to \mathrm{FC}(256) \to \mathrm{FC}(512) \to \mathbf{p}^{(i,l)} \tag{11}$$

The reconstruction error in this case is the binary cross-entropy: $\log p_\theta(\mathbf{b}^{(i)} \mid \mathbf{a}^{(i)}, \mathbf{z}^{(i,l)}) = -\sum \mathbf{b}^{(i)} \log \mathbf{p}^{(i,l)} + (1 - \mathbf{b}^{(i)}) \log(1 - \mathbf{p}^{(i,l)})$. For real-valued data such as Yale facial images and RAVDESS log-mel spectrograms, we let $p_\theta(\mathbf{b} \mid \mathbf{a}^{(i)}, \mathbf{z}^{(i,l)})$ be a multivariate Gaussian distribution with a fixed diagonal covariance $\mathcal{N}(\mathbf{b}; \boldsymbol{\mu}^{(i,l)}, \boldsymbol{\sigma}^2 \mathbf{I})$ where $\boldsymbol{\mu}^{(i,l)}$ is computed from MLPs $f_\theta^p(\mathbf{a}^{(i)}, \mathbf{z}^{(i)})$:

$$f_\theta^p : [\mathbf{a}^{(i)}, \mathbf{z}^{(i,l)}] \to \mathrm{FC}(512) \to \mathrm{FC}(256) \to \mathrm{FC}(512) \to \boldsymbol{\mu}^{(i,l)}, \tag{12}$$

The reconstruction error in this case is the mean squared error: $\log p_\theta(\mathbf{b}^{(i)} \mid \mathbf{a}^{(i)}, \mathbf{z}^{(i,l)}) = \frac{-1}{\boldsymbol{\sigma}^2} \|\mathbf{b}^{(i)} - \boldsymbol{\mu}^{(i,l)}\|^2 + \mathrm{const}$. We experimented with both continuous and discrete (categorical) latent variable $\mathbf{z}$. For a continuous $\mathbf{z}$, we let the prior $p_\theta(\mathbf{z})$ be a bivariate normal distribution and the approximated posterior $q_\phi(\mathbf{z} \mid \mathbf{a}'^{(i)}, \mathbf{b}'^{(i)})$ be a bivariate Gaussian distribution with a diagonal covariance $\mathcal{N}(\mathbf{z}; \boldsymbol{\mu}^{(i)}, (\boldsymbol{\sigma}^{(i)})^2 \mathbf{I})$ where $\boldsymbol{\mu}^{(i)}$ and $\boldsymbol{\sigma}^{(i)}$ are computed from MLPs $f_\phi^q(\mathbf{a}'^{(i)}, \mathbf{b}'^{(i)})$:

$$f_\phi^q : [\mathbf{a}'^{(i)}, \mathbf{b}'^{(i)}] \to \mathrm{FC}(512) \to \mathrm{FC}(256) \to [\boldsymbol{\mu}^{(i)}, \boldsymbol{\sigma}^{(i)}] \tag{13}$$

In this case the latent variable samples are drawn as: $\mathbf{z}^{(i,l)} = \boldsymbol{\mu}^{(i)} + \boldsymbol{\sigma}^{(i)} \odot \boldsymbol{\epsilon}^{(i,l)}, \ \boldsymbol{\epsilon}^{(i,l)} \sim \mathcal{N}(\mathbf{0}, \mathbf{I})$. For a discrete $\mathbf{z}$, we used two categorical r.v., $\mathbf{z} = [\mathbf{z}_1, \mathbf{z}_2]$, each having a uniform prior over five categories and let $q_\phi(\mathbf{z} \mid \mathbf{a}'^{(i)}, \mathbf{b}'^{(i)})$ represents two categorical r.v. reparameterized with Gumbel-Softmax distributions whose class probabilities $\boldsymbol{\pi}_1 = (\pi_{11}, \dots, \pi_{15})$ and $\boldsymbol{\pi}_2 = (\pi_{21}, \dots, \pi_{25})$ are computed from MLPs $f_\phi^q(\mathbf{a}'^{(i)}, \mathbf{b}'^{(i)})$ (Jang et al., 2017; Maddison et al., 2017):

$$f_\theta^p : [\mathbf{a}'^{(i)}, \mathbf{b}'^{(i)}] \to \mathrm{FC}(512) \to \mathrm{FC}(256) \to [\boldsymbol{\pi}_1, \boldsymbol{\pi}_2] \tag{14}$$

In this case the latent variable samples are drawn as: $\mathbf{z}_j^{(i,l)} = \mathrm{softmax}\big((\boldsymbol{\epsilon}^{(i,l)} + \log \boldsymbol{\pi}_j)/\tau\big), \ j = 1, 2$ where $\boldsymbol{\epsilon}^{(i,l)} \in \mathbb{R}^5$ are $i.i.d.$ samples drawn from a $\mathrm{Gumbel}(0,1)$ distribution and the softmax temperture $\tau$ controls the "smoothness" of the samples.

For MNIST, Omniglot, and Yale relational learning tasks, we used random image rotation for RPDA, i.e., $D = \{ (\mathrm{rot}(\mathbf{a}, r), \mathrm{rot}(\mathbf{b}, r)) \mid r \in [0, 360) \}$. For time-series RAVDESS speech waveforms, we used random time delay and amplitude scaling for RPDA, i.e., $D = \{ (\alpha \mathbf{a}[t - \tau], \alpha \mathbf{b}[t - \tau]) \mid \alpha \in \mathbb{R}, \tau \in \mathbb{Z} \}$. Additionally, since we are only interested in learning an *undirected* relative relationship

changes in Yale and RAVDESS relational learning tasks (i.e., $p_\theta(\mathbf{z} \mid \mathbf{a}, \mathbf{b}) = p_\theta(\mathbf{z} \mid \mathbf{b}, \mathbf{a})$), we augment RPDA functions with random swapping operations:

$$\mathrm{swap}(\mathbf{a}', \mathbf{b}') = \begin{cases} (\mathbf{a}', \mathbf{b}'), & p = 0.5 \\ (\mathbf{b}', \mathbf{a}'), & p = 0.5 \end{cases}$$

Finally, the learning objective $\widetilde{\mathcal{L}}_{\mathrm{RPDA}}^{(i)}$ in Eq. 10 can be constructed once $p_\theta(\mathbf{b} \mid \mathbf{a}^{(i)}, \mathbf{z}^{(i,l)})$, $p_\theta(\mathbf{z})$, and $q_\phi(\mathbf{z} \mid \mathbf{a}'^{(i)}, \mathbf{b}'^{(i)})$ are defined. For example, when we adopt a discrete latent variable $\mathbf{z} = [\mathbf{z}_1, \mathbf{z}_2]$ for MNIST and Omniglot relational learning tasks (Sec. 5.1 and 5.2), we can derive $\widetilde{\mathcal{L}}_{\mathrm{RPDA}}^{(i)}$ (with $L = 1$) as:

$$\widetilde{\mathcal{L}}_{\mathrm{RPDA}}^{(i)} = -\sum \mathbf{b}^{(i)} \log \mathbf{p}^{(i,1)} + (1 - \mathbf{b}^{(i)}) \log(1 - \mathbf{p}^{(i,1)}) - \sum \boldsymbol{\pi}_j \log \boldsymbol{\pi}_j + \mathrm{const} \qquad (15)$$

where $(\mathbf{a}'^{(i)}, \mathbf{b}'^{(i)}) = (\mathrm{rot}(\mathbf{a}, r^{(i)}), \mathrm{rot}(\mathbf{b}, r^{(i)}))$, $r^{(i)} \sim \mathcal{U}([0, 360))$, $[\boldsymbol{\pi}_1, \boldsymbol{\pi}_2] = f_\phi^q(\mathbf{a}'^{(i)}, \mathbf{b}'^{(i)})$, $\mathbf{z}_j^{(i,1)} = \mathrm{softmax}((\boldsymbol{\epsilon}^{(i,1)} + \log \boldsymbol{\pi}_j)/\tau)$, $\boldsymbol{\epsilon}^{(i,1)} \sim \mathrm{Gumbel}(0,1)$, $j = 1, 2$, and $\mathbf{p}^{(i,1)} = f_\theta^p(\mathbf{a}^{(i)}, [\mathbf{z}_1^{(i,1)}, \mathbf{z}_2^{(i,1)}])$. Similarly, $\widetilde{\mathcal{L}}_{\mathrm{RPDA}}^{(i)}$ (with $L = 1$) for a continuous latent variable $\mathbf{z}$ can be derived as:

$$\begin{aligned} \widetilde{\mathcal{L}}_{\mathrm{RPDA}}^{(i)} = &-\sum \mathbf{b}^{(i)} \log \mathbf{p}^{(i,1)} + (1 - \mathbf{b}^{(i)}) \log(1 - \mathbf{p}^{(i,1)}) + \log \mathcal{N}(\mathbf{z}^{(i,1)}; \mathbf{0}, \mathbf{I}) \\ &- \log \mathcal{N}(\mathbf{z}^{(i,1)}; \boldsymbol{\mu}^{(i)}, (\boldsymbol{\sigma}^{(i)})^2 \mathbf{I}), \end{aligned} \qquad (16)$$

where $(\mathbf{a}'^{(i)}, \mathbf{b}'^{(i)}) = (\mathrm{rot}(\mathbf{a}, r^{(i)}), \mathrm{rot}(\mathbf{b}, r^{(i)}))$, $r^{(i)} \sim \mathcal{U}([0, 360))$, $[\boldsymbol{\mu}^{(i)}, \boldsymbol{\sigma}^{(i)}] = f_\phi^q(\mathbf{a}'^{(i)}, \mathbf{b}'^{(i)})$, $\mathbf{z}^{(i,1)} = \boldsymbol{\mu}^{(i)} + \boldsymbol{\sigma}^{(i)} \odot \boldsymbol{\epsilon}^{(i,1)}$, $\boldsymbol{\epsilon}^{(i,1)} \sim \mathcal{N}(\mathbf{0}, \mathbf{I})$, and $\mathbf{p}^{(i,1)} = f_\theta^p(\mathbf{a}^{(i)}, \mathbf{z}^{(i,1)})$. For Yale and RAVDESS relational learning tasks (Sec. 5.3), we adopted a continuous latent variable $\mathbf{z}$ with $\boldsymbol{\sigma} = 0.1$ and $\widetilde{\mathcal{L}}_{\mathrm{RPDA}}^{(i)}$ (with $L = 1$) can be derived as:

$$\widetilde{\mathcal{L}}_{\mathrm{RPDA}}^{(i)} = \frac{-1}{0.02} \|\mathbf{b}^{(i)} - \boldsymbol{\mu}^{(i,l)}\|^2 + \log \mathcal{N}(\mathbf{z}^{(i,1)}; \mathbf{0}, \mathbf{I}) - \log \mathcal{N}(\mathbf{z}^{(i,1)}; \boldsymbol{\mu}^{(i)}, (\boldsymbol{\sigma}^{(i)})^2 \mathbf{I}) + \mathrm{const} \qquad (17)$$

where $(\mathbf{a}'^{(i)}, \mathbf{b}'^{(i)}) = \mathrm{swap}(\mathrm{rot}(\mathbf{a}, r^{(i)}), \mathrm{rot}(\mathbf{b}, r^{(i)}))$, $r^{(i)} \sim \mathcal{U}([0, 360))$, $[\boldsymbol{\mu}^{(i)}, \boldsymbol{\sigma}^{(i)}] = f_\phi^q(\mathbf{a}'^{(i)}, \mathbf{b}'^{(i)})$, $\mathbf{z}^{(i,1)} = \boldsymbol{\mu}^{(i)} + \boldsymbol{\sigma}^{(i)} \odot \boldsymbol{\epsilon}^{(i,1)}$, $\boldsymbol{\epsilon}^{(i,1)} \sim \mathcal{N}(\mathbf{0}, \mathbf{I})$, and $\boldsymbol{\mu}^{(i,l)} = f_\theta^p(\mathbf{a}^{(i)}, \mathbf{z}^{(i,1)})$.

All MLPs with parameters $\theta$ and $\phi$ were jointly trained for 300k iterations (without batch-normalization, weight decay, nor dropout) to maximize $\widetilde{\mathcal{L}}_{\mathrm{RPDA}}^{(i)}$ in Eq. (4) with using Adam optimizer (learning rate=0.0004, $\beta_1$=0.9, $\beta_1$=0.999) (Kingma & Ba, 2015). Minibatches of size M=100 were used. We anneal the learning rate (0.0004 base learning rate) with step decay (factor of 0.5 every 100k iterations). When Gumbel-Softmax distributions is used, we anneal the softmax temperature $\tau$ from 1.0 to 0.5 with exponential decay (decay rate=0.00005).

### E.2 BASELINE METHODS IMPLEMENTATION

**STN-affine** and **STN-rotate** (Jaderberg et al., 2015; Dong et al., 2017) minimize the difference between $\mathbf{b}^{(i)}$ and a geometric transformation of $\mathbf{a}^{(i)}$ (affine transformation for STN-affine and rotation transformation for STN-rotate) by using a spatial transformer with a localization network that takes both $\mathbf{a}^{(i)}, \mathbf{b}^{(i)}$ as input; during evaluation, we perform clustering on the output of the trained localization network. The localization network in STN-affine takes both $\mathbf{a}^{(i)}, \mathbf{b}^{(i)}$ as input and outputs an affine transformation matrix:

$$[\mathbf{a}^{(i)}, \mathbf{b}^{(i)}] \to \mathrm{FC}(512) \to \mathrm{FC}(256) \to \mathrm{FC}(256) \to \boldsymbol{A}_\theta = \begin{bmatrix} \theta_{11} & \theta_{12} & \theta_{13} \\ \theta_{21} & \theta_{22} & \theta_{23} \end{bmatrix}.$$

In STN-rotate, we further restrict STN-affine to only allow rotation transformation and the localization network in this case only outputs the rotation angle $\Delta\theta$:

$$[\mathbf{a}^{(i)}, \mathbf{b}^{(i)}] \to \mathrm{FC}(512) \to \mathrm{FC}(256) \to \mathrm{FC}(256) \to \Delta\theta, \quad \boldsymbol{A}_\theta = \begin{bmatrix} \cos(\Delta\theta) & -\sin(\Delta\theta) & 0 \\ \sin(\Delta\theta) & \cos(\Delta\theta) & 0 \end{bmatrix}.$$

**VAE, AAE, GMVAE** all trained directly using $(\mathbf{a}^{(i)}, \mathbf{b}^{(i)})$ and we perform clustering on their learned latent space. We selected dim($\mathbf{z}$)=10 via cross-validation. We follow the implementation of (Kingma

& Welling, 2014; Makhzani et al., 2015; Dilokthanakul et al., 2017) since they also experimented with MNIST or Omniglot dataset. However, for a fair comparison, we also experimented with the following encoder/decoder architecture and report the best results:

$$[\mathbf{a}^{(i)}, \mathbf{b}^{(i)}] \rightarrow \text{FC}(512) \rightarrow \text{FC}(256) \rightarrow \mathbf{z}^{(i)} \sim q(\mathbf{z}^{(i)}) \rightarrow \text{FC}(256) \rightarrow \text{FC}(512) \rightarrow [\mathbf{y}_{\mathbf{a}^{(i)}}, \mathbf{y}_{\mathbf{b}^{(i)}}].$$

Our **VAE-contrastive** implementation is similar to Chen et al. (2020) in concept. We added an additional latent loss term to the VAE loss function that aim to minimize the difference between the latent representation from different rotation augmented image pairs, i.e., $\alpha \|\mathbf{z}^{(i)} - \mathbf{z}'^{(i)}\|_2$ (also experimented with cosine similarity) where $\mathbf{z}^{(i)} = Enc([\mathbf{a}^{(i)}, \mathbf{b}^{(i)}])$, $\mathbf{z}'^{(i)} = Enc([\mathbf{a}'^{(i)}, \mathbf{b}'^{(i)}])$ and we cross-validate $\alpha$ from 0.1 to 10.

**BYOL** learns a joint image represenation $\mathbf{z}$ for $(\mathbf{a}^{(i)}, \mathbf{b}^{(i)})$, on which the clustering is applied during evaluation. We selected dim($\mathbf{z}$)=10 via cross-validation.

**Vec-arithmetic** first trains an autoencoder on individual $\mathbf{a}^{(i)}, \mathbf{b}^{(i)}$ and then used the trained encoder to compute a latent vector representation of $(\mathbf{a}^{(i)}, \mathbf{b}^{(i)})$ through $Enc(\mathbf{b}^{(i)}) - Enc(\mathbf{a}^{(i)})$, on which the clustering is applied during evaluation. We selected dim($\mathbf{z}$)=10 via cross-validation.

**LICM** (Evtimova, 2017) trained a committee of 5 experts to learn a set of independent mechanisms that generates $\mathbf{b}^{(i)}$ from the canonical $\mathbf{a}^{(i)}$, and used the trained discriminator during evaluation to select the winning expert (as a categorical variable).

**NRI** infer the interaction between $\mathbf{a}^{(i)}$ and $\mathbf{b}^{(i)}$ and used the NRI-Encoder to predict the relationship type (as a categorical variable). We follow the implementation of (Kipf et al., 2018; Fetaya, 2019) where we used the NRI-Encoder to infer the relationhip between $\mathbf{a}^{(i)}$ and $\mathbf{b}^{(i)}$; the inferred relationship $\mathbf{z}^{(i)}$, together with $(\mathbf{a}^{(i)}, \mathbf{b}^{(i)})$, are then feed into the NRI-Decoder to predict $\mathbf{y}_{\mathbf{a}^{(i)}}$ and $\mathbf{y}_{\mathbf{b}^{(i)}}$ which are reconstructions for $\mathbf{a}^{(i)}$ and $\mathbf{b}^{(i)}$, respectively. To account for the high-dimensional input data, we used two MLPs (one for each $\mathbf{a}^{(i)}$ and $\mathbf{b}^{(i)}$) to extract lower dimensional feature vectors $\boldsymbol{x}_{\mathbf{a}^{(i)}} \in \mathbb{R}^{10}$ and $\boldsymbol{x}_{\mathbf{b}^{(i)}} \in \mathbb{R}^{10}$ that serve as input to NRI-Encoder:

$$\mathbf{a}^{(i)} \rightarrow \text{FC}(512) \rightarrow \text{FC}(256) \rightarrow \mathbf{x}_{\mathbf{a}^{(i)}}, \quad \mathbf{b}^{(i)} \rightarrow \text{FC}(512) \rightarrow \text{FC}(256) \rightarrow \mathbf{x}_{\mathbf{b}^{(i)}}.$$

Similarly, we insert two MLPs (with the same architecture as above) in front of NRI-Decoder to extract data feature vectors and append two MLPs after NRI-Decoder to convert the predicted feature vectors back to the high-dimensional data space:

$$\mathbf{x}_{\mathbf{a}^{(i)}} \rightarrow \text{FC}(256) \rightarrow \text{FC}(512) \rightarrow \mathbf{y}_{\mathbf{b}^{(i)}}, \quad \mathbf{x}_{\mathbf{b}^{(i)}} \rightarrow \text{FC}(512) \rightarrow \text{FC}(256) \rightarrow \mathbf{y}_{\mathbf{b}^{(i)}}.$$

All MLPs are trained jointly with NRI encoder and decoder.

For our final baseline comparison, we present results from applying InfoGAN to the MNIST relational learning task in Sec. 5.1 (Chen et al., 2016). InfoGAN has demonstrated its ability to learn disentangled representations (represented by structured latent codes $c_1, c_2, ..., c_L$) through generative modelling. Although inferring latent codes for a given data point is a non-trivial task for InfoGAN, we can examine the learned latent representation by manipulating the latent codes and visually inspect the generated random samples. We modeled the latent codes with one categorical code $c_1 \sim \text{Cat}(K = 10, p = 0.1)$ and two continuous codes $c_2, c_3 \sim \text{Unif}(-1, 1)$. Figure 7 shows examples of generated images from manipulating the latent codes and it is clear that none of the latent codes distinctively capture the full range of relative rotational relationships.

## F  ADDITIONAL EXPERIMENTAL RESULTS AND ANALYSIS

### F.1  MNIST AND OMNIGLOT RELATIONAL LEARNING RESULTS ANALYSIS

At first glance, the results in Fig. 3c resemble that of style-transfer, but they are fundamentally different: in style-transfer, the image $\mathbf{b}^{(r,c)}$ is generated by applying the *style* of $\mathbf{b}_s^{(r)}$ to the *content* of $\mathbf{a}^{(c)}$, whereas VRL generates image $\mathbf{b}^{(r,c)}$ by applying the *relational property* of $(\mathbf{a}_s, \mathbf{b}_s^{(r)})$ to the image $\mathbf{a}^{(c)}$. It is evident from Fig. 3c that predicted images $\mathbf{b}^{(r,c)}$ do not share similar *style* to $\mathbf{b}_s^{(r)}$, but rather the same relative rotational relationship w.r.t. $\mathbf{a}^{(c)}$ and $\mathbf{a}_s$.

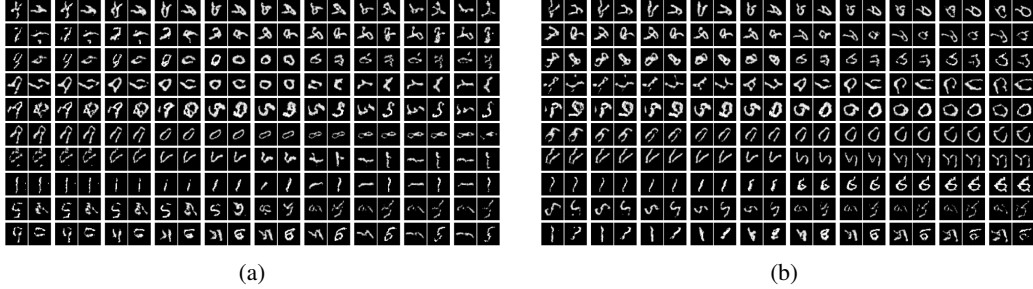

|              |              |
| :----------: | :----------: |
| (a)          | (b)          |

Figure 7: Manipulating latent codes of InfoGAN on MNIST where each row represents random samples from varying continuous latent code $c_2$ in (a) and $c_3$ in (b) while other latent codes and noise are fixed; different rows correspond to different categorical code $c_1 = 1 \dots 10$.

A close inspection of the results in Table 1 shows that the seemingly simple relational learning tasks is in fact very challenging even for specialized methods such as STN-affine and STN-rotate that only learn about spatial transformations. Comparing VRL with VAE, we can see that VRL achieved better performance than VAE with more efficient learning—VAE were only able to solve the simpler 3-relationship tasks by adopting a high dimensional latent space; in contrast, VRL were able to solve all tasks with a compact 2-D latent space. Further comparing VAE with GMVAE and AAE shows that regularizing the latent space, as done in AAE and GMVAE, can degrade the performance of relational learning. For contrastive self-supervised learning methods like VAE-contrastive and BYOL there is no way to enforce the learned data representation be independent of absolute property since different augmented image still share large part of their absolute property (e.g., digit representation). For example, it is entirely possible for VAE-contrastive and BYOL to tightly cluster images based on their digit representation (absolute property); this will lead to a small loss function value but makes the subsequent relational discrimination task more challenging since each digit representation can have all possible relative rotation relationships. LICM failed all relational learning tasks due to the fact that the transformed and canonical distributions overlaps completely (both $\mathbf{a}^{(i)}$ and $\mathbf{b}^{(i)}$ are rotations of MNIST or Omniglot images) and it is not possible for a committee of experts to distinguish the two. NRI's inability to solve the relational learning tasks can be attributed to the fact that it is designed to learn in a dynamical systems, but more importantly, GNN is not guaranteed to learn an independent relative relationship.

In summary, we argue that a major challenge for applying existing methods to relational learning problem is that they learn a single representation that encodes both relational and absolute properties and it is difficult to dissect the relational property from the learned representation.

### F.2  YALE AND RAVDESS RELATIONAL MAPPING RESULTS

Figure 8 shows examples of relational mappings predicted by the same set of trained VRL models from Sec. 5.3.

### F.3  RELATIONAL LEARNING WITH MULTIPLE RELATIONSHIPS

Here, we setup a more complex relational learning task that includes both relative rotational and scaling relationships. We constructed a paired MNIST dataset $\mathbf{X}^{M_{10}} = \{ (\mathbf{a}^{(i)}, \mathbf{b}^{(i)}) \mid i \in [1..N] \}$ where each $\mathbf{a}^{(i)}$ is a randomly rotated and scaled (by a factor $\times 0.66$ or $\times 1$) MNIST image and $\mathbf{b}^{(i)}$ is another random rotation (by $0°$, $72°$, $144°$, $216°$, or $288°$) and scaling (by $\times 1$ or $\times 1.5$) of $\mathbf{a}^{(i)}$. Note that there are a total of 10 decoupled relative relationships between $(\mathbf{a}^{(i)}, \mathbf{b}^{(i)})$ in $\mathbf{X}^{M_{10}}$ (combinations of 5 rotational and 2 scaling transformations). We trained VRL on $\mathbf{X}^{M_{10}}$ following the same training procedure as described in Appendix E but with larger MLPs for $f_\theta^p(\mathbf{a}^{(i)}, \mathbf{z}^{(i)})$ and

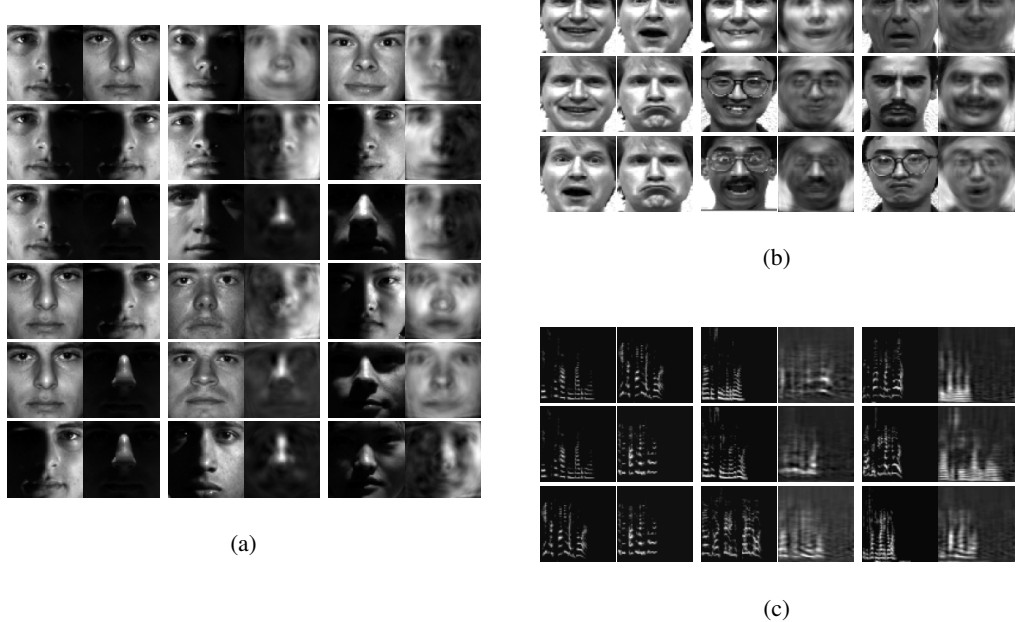

(a)

(b)

(c)

Figure 8: Examples of VRL relational mapping for facial images and speech waveforms; each pair of images shows VRL relational mapping predictions based on the relational property inferred from the first two images in the same row; (a) Relative facial illumination condition changes; (b) Relative facial expression changes; (c) Relative speech emotion changes.

$f_\phi^q(\mathbf{a}'^{(i)}, \mathbf{b}'^{(i)})$ (cf. Eq. 11, 13) to account for the increased problem complexity:

$$
\begin{aligned}
f_\theta^p &: [\mathbf{a}^{(i)}, \mathbf{z}^{(i,l)}] \to \mathrm{FC}(1024) \to \mathrm{FC}(512) \to \mathrm{FC}(512) \to \mathbf{p}^{(i,l)} \\
f_\phi^q &: [\mathbf{a}'^{(i)}, \mathbf{b}'^{(i)}] \to \mathrm{FC}(1024) \to \mathrm{FC}(512) \to [\boldsymbol{\mu}^{(i)}, \boldsymbol{\sigma}^{(i)}]
\end{aligned}
\tag{18}
$$

The inference result is shown in Fig. 9a, where we can see that the approximated posterior accurately cluster (discriminate) data with the same (different) relative relationship together (apart). Examples of images predicted by direct sampling in the latent space are shown in Fig. 9b.

### F.4 RELATIONAL LEARNING WITH CONTINUOUS RELATIONSHIPS

Lastly, we present an example with a *continuous* relational property. Based on the MNIST dataset, we constructed a paired dataset $\mathbf{X}^{M_c} = \{\, (\mathbf{a}^{(i)}, \mathbf{b}^{(i)}) \mid i \in [1..N] \,\}$ where both $\mathbf{a}^{(i)}$ and $\mathbf{b}^{(i)}$ are random rotation of the same MNIST image. In this case, there is a continuous (and decoupled) relative rotational relationship between $(\mathbf{a}^{(i)}, \mathbf{b}^{(i)})$. We trained VRL on $\mathbf{X}^{M_c}$ following the same training procedure as described in Appendix E but used convolutional neural networks (CNNs) to capture the continuous relationship. We approximate $p_\theta(\mathbf{b} \mid \mathbf{a}^{(i)}, \mathbf{z}^{(i,l)})$ with an autoencoder-like neural network $f_\theta^p(\mathbf{a}^{(i)}, \mathbf{z}^{(i)}) = f_\theta^{\mathrm{dec}}\left(f_\theta^{\mathrm{enc}}(\mathbf{a}^{(i)}), \mathbf{z}^{(i,l)}\right)$:

$$
\begin{aligned}
f_\theta^{\mathrm{enc}} &: \mathbf{a}^{(i)} \to \mathrm{Conv}(3\text{x}3\text{x}8) \to \mathrm{Conv}(3\text{x}3\text{x}32) \to \mathrm{Conv}(3\text{x}3\text{x}128) \to \mathrm{FC}(20) \to \mathbf{h}^{(i)} \in \mathbb{R}^{20} \\
f_\theta^{\mathrm{dec}} &: [\mathbf{h}^{(i)} \in \mathbb{R}^{20}, \mathbf{z}^{(i,l)}] \to \mathrm{FC} \to \mathrm{Conv}^T(3\text{x}3\text{x}128) \to \mathrm{Conv}^T(3\text{x}3\text{x}32) \to \mathrm{Conv}^T(3\text{x}3\text{x}8) \\
&\quad \to \mathrm{Conv}(1\text{x}1\text{x}1) \xrightarrow{\mathrm{Sigmoid}} [0,1]^{\dim(\mathcal{B})},
\end{aligned}
$$

where $\mathrm{Conv}(\cdot)$ is a strided (stride 2) convolutional layer and $\mathrm{Conv}^T(\cdot)$ is a transposed convolutional layer. We used batch-normalization after each layer and rectified linear non-linearities. We represent the approximated posterior $q_\phi(\mathbf{z} \mid \mathbf{a}'^{(i)}, \mathbf{b}'^{(i)})$ (with bivariate normal distribution as prior) using:

$$
f_\phi^q : [\mathbf{a}'^{(i)}, \mathbf{b}'^{(i)}] \to \mathrm{Conv}(3\text{x}3\text{x}8) \to \mathrm{Conv}(3\text{x}3\text{x}32) \to \mathrm{Conv}(3\text{x}3\text{x}128) \to \mathrm{FC}(4) \to [\boldsymbol{\mu}^{(i)}, \boldsymbol{\sigma}^{(i)}].
$$

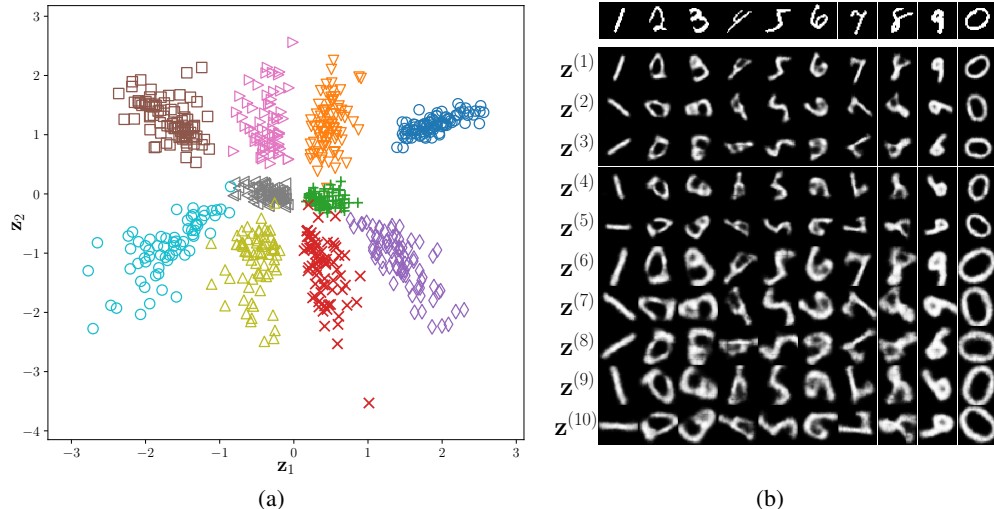

(a)          (b)

Figure 9: Learning multiple relative relationship with VRL: (a) Scatter plot of 2-D relational property of $\mathbf{X}^{M_{10}}$ inferred by VRL (relative relationship labels: $\bigcirc$(blue) : $0°$, $\triangledown$ : $72°$, $+$ : $144°$, $\times$ : $216°$, $\diamondsuit$ : $288°$, $\square$ : $0°$, $\times 1.5$, $\triangleright$ : $72°$, $\times 1.5$, $\triangleleft$ : $144°$, $\times 1.5$, $\triangle$ : $216°$, $\times 1.5$, $\bigcirc$(cyan) : $288°$, $\times 1.5$); (b) Images predicted from sampled latent variables (sampling the centroid of each cluster in (a): "$\bigcirc$"(blue)$\rightarrow \mathbf{z}^{(1)}$, "$\triangledown$"$\rightarrow \mathbf{z}^{(2)}$,"$+$"$\rightarrow \mathbf{z}^{(3)}$, "$\times$"$\rightarrow \mathbf{z}^{(4)}$, "$\diamondsuit$"$\rightarrow \mathbf{z}^{(5)}$, "$\square$"$\rightarrow \mathbf{z}^{(6)}$, "$\triangleright$"$\rightarrow \mathbf{z}^{(7)}$, "$\triangleleft$"$\rightarrow$ $\mathbf{z}^{(8)}$, "$\triangle$"$\rightarrow \mathbf{z}^{(9)}$, "$\bigcirc$"(cyan)$\rightarrow \mathbf{z}^{(10)}$).

A scatter plot of the relational property inferred by a trained posterior is shown in Fig. 10a, and examples of images predicted by direct sampling in the latent space (denoted by markers "$\times$" in Fig. 10a) are shown in Fig. 10b. From Fig. 10 we can see that VRL learned an independent relational

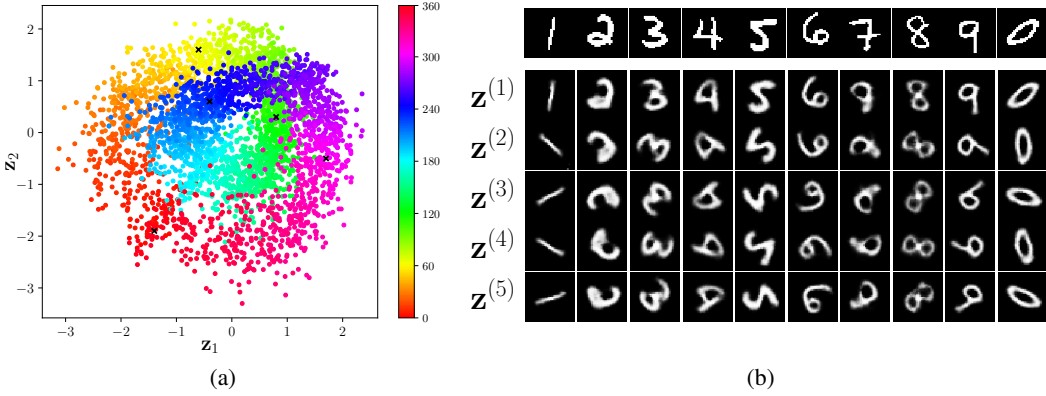

(a)          (b)

Figure 10: Learning continuous relative relationship with VRL: (a) Scatter plot of 2-D relational property of $\mathbf{X}^{M_c}$ inferred by VRL, each point is color-coded (best viewed in color) by the degrees of relative rotation between the corresponding data point; (b) Images predicted from sampled latent variables (denoted by markers "$\times$" in (a)).

property that encodes a continuous relative rotational relationship; however, there is a small region in Fig. 10a with overlapping relational property that leads to an ambiguous relationship interpretation ($120°$ vs. $240°$). This ambiguity is likely caused by compressing the relative relationship down to a 2-D latent space, $\mathbf{z} \in \mathbb{R}^2$, and motivates us to adopt a higher-dimensional latent space, e.g., $\mathbf{z} \in \mathbb{R}^3$. Figure 11 shows inference result from repeating the previous experiment but with adopting $\mathbf{z} \in \mathbb{R}^3$. We can see that VRL learned a three-dimensional relational property that unambiguously represents the underlying continuous relative rotational relationship.

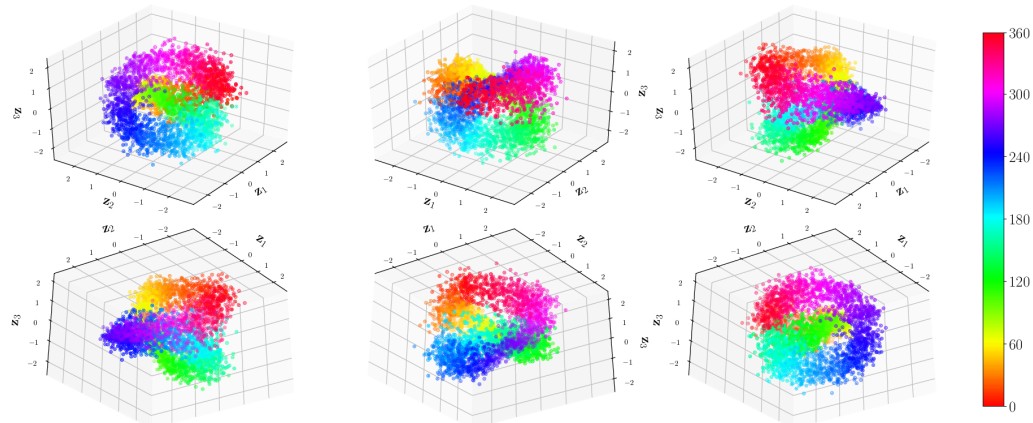

Figure 11: Scatter plot of 3-D relational property of $\mathbf{X}^{M_c}$ inferred by VRL (with $\mathbf{z} \in \mathbb{R}^3$); each plot shows a different vantage point of the 3-D scatter plot, and each point is color-coded (best viewed in color) by the degrees of relative rotation between the corresponding data point.

