# OpenReview forum: "Relational Learning with Variational Bayes"
_ICLR.cc/2022/Conference — ICLR 2022 Poster_

### Official Review · Reviewer_Znwn · 2021-10-24

**Correctness:** 3
**Technical Novelty And Significance:** 3
**Empirical Novelty And Significance:** 3
**Recommendation:** 5
**Confidence:** 4

**Main Review:**

Strengths:
- the idea of learning relations solely from existing relations is interesting
- the theoretical background is simple and well defined

Weaknesses:
- datasets are only images
- from the authors' example (happy-sad faces), it seems absolute properties are indirectly being used
- how does this work differ from others that work on complex networks, trying to extract motifs from graphs?



**Summary Of The Paper:**

This paper proposes a relational learning method based on variational Bayes. The main idea is to learn relations among objects independently of each object's own properties. The model theory is discusses where authors define the problem and discuss about solutions for some limitations of the model when a relation between objects A and B can be found by only looking at one of the objects own properties.
The paper is quite well written. Algorithms are provided, but not the code. As far as I understood, results are obtained on "benchmark" datasets, where some synthetic versions of these datasets are used to create the relations between the instances of the original dataset and of the synthetic.

**Summary Of The Review:**

Overall, I liked the idea of formulating the problem as "learning relations from relations". However, there are other approaches that are not mentioned here which try to find "motifs" in graphs or perform link prediction, where the objects (without their properties) are represented as nodes, relations are in the edges and new links are learned from the original graph.

From your example (sad-happy faces), it seems features of each image are being used to learn new relations. Why did you choose images for training? I would think that a tabular dataset (specially in areas such as health) could benefit much from your approach.

References are outdated. The last year for your refs is 2019. There has been a whole body of work regarding relational learning and methods developed in 2020 and 2021.

---

> ### Author Response · Authors · 2021-11-10
> **Author response to Reviewer Znwn**
>
> Dear Reviewer Znwn:
>
> Thank you for the detailed review and comments. Please find our response to your concerns/questions below:
>
> > datasets are only images
>
> > I would think that a tabular dataset (specially in areas such as health) could benefit much from your approach.
>
> We use image examples for ease of illustration as there is no existing benchmark relational learning tasks for unsupervised methods. As an initial investigation into this overlooked problem, our goal is to design a controlled experiments that allow us to rigorously and quantitatively test various relational learning tasks as well as providing a common problem settings where we can compare with a wide range of existing unsupervised methods. We believe our empirical results achieved these goals. Relational learning certainly can be applied to tabular dataset such as public health record but these results would rely on subjective interpretation that can only generate anecdotal evidence and not conducive for rigorous quantitative evaluation; furthermore, it would require significant effort from domain expert to build a large-scale training dataset. For the above reason, while we agree this is a very good suggestion, we believe this deserves dedicated study and is outside the scope of our current work.
>
> > from the authors' example (happy-sad faces), it seems absolute properties are indirectly being used
>
> > From your example (sad-happy faces), it seems features of each image are being used to learn new relations.
>
> Figure 5 (b) shows that VRL learns to distinguish undirected relationships "happy-sad" (consisting all (happy, sad) and (sad, happy) image pairs) from "happy-surprised" (consisting all (happy, surprised) and (surprised, happy) image pairs) and "surprised-sad"(consisting all (surprised, sad) and (sad, surprised) image pairs). Results in Fig. 5 (b) cannot be obtained by looking at the absolute property (e.g. subject identity, expression, etc.) of individual image alone. We stress again, in our work we define absolute property as information that characterizes the marginal distribution $p(a)$ and $p(b)$. We require relational property $z$ to be independent of $a$ and $b$, i.e., $p(z|a)=p(z|b)=p(z)$, but it will necessarily draw information from observing both $a$ and $b$, i.e., $p(z|a,b) \neq p(z)$.
>
> > how does this work differ from others that work on complex networks, trying to extract motifs from graphs?
>
> > However, there are other approaches that are not mentioned here which try to find "motifs" in graphs or perform link prediction, where the objects (without their properties) are represented as nodes, relations are in the edges and new links are learned from the original graph.
>
> > References are outdated. The last year for your refs is 2019. There has been a whole body of work regarding relational learning and methods developed in 2020 and 2021.
>
> Many of the methods we compared against in Table 1 are directly based on graphical model or graph network (e.g., VAE, NRI) other methods have equivalent graphical model interpretation. We exclude some other popular link prediction methods in our discussion since they do not fit our problem setting for obvious reasons. For example, link prediction based on node attributes similarity directly contradicts our goal of learning a relational property that is independent of absolute property (node attributes). Other popular link prediction methods based on matrix factorization typically require some supervised dataset (e.g., ratings matrices) for training which we do not have. Each method in Table 1 represents a unique class of algorithms and, collectively, we believe they are a good representation of current state-of-the-art unsupervised methods. We welcome any suggestion that points us to other relevant work we may have missed.
>
> We hope this answers your question and look forward to additional discussion.

---

> > ### Comment · Reviewer_Znwn · 2021-11-27
> > **Response to authors's rebuttal**
> >
> > Dear authors, many thanks for your careful response. Regarding datasets, while I agree that you should use a controlled experiment to evaluate your method, I am not convinced that you can not find other relational data that you could use to illustrate how the proposed method would behave in non-synthetic data. There are plenty of datasets that are available from social networks and alike.
> > Regarding other methods in the literature, there is a whole body of work in Statistical Relational Learning (SRL) with models like MLNs, DeepProbLog, ProbFOIL, SlipCover, PGMs etc or even works based on predicate invention and neurosymbolic approaches. Some of them are used for supervised learning, but, nevertheless, they learn new relations from primitive data or from other relations, and could be used for unsupervised learning.
> > And regarding my question about the dates of your references? Aren't there any works more recent than 2019 in this area?

---

> > > ### Author Response · Authors · 2021-11-27
> > > **Author's additional response**
> > >
> > > Dear Reviewer Znwn:
> > >
> > > Thank you for sharing with us your additional concerns. Please see our response below:
> > >
> > > > I am not convinced that you can not find other relational data that you could use to illustrate how the proposed method would behave in non-synthetic data
> > >
> > > Please see the seminal work by Christie & Gentner (2010, "Where hypotheses come from: learning new relations by structural alignment") and Köhler (1929) where the authors conducted some of the most important experiments in the field of relational learning by using synthetic datasets (which also motivates our experimental design in this work). Just as the authors in those landmark papers used "synthetic data" to study human's unique learning behavior, we used synthetic examples to highlight key properties/behaviors of our proposed unsupervised learning method. Finally, while our examples are synthetic, the data itself are real-world images (MNIST, facial images).
> > >
> > > > There are plenty of datasets that are available from social networks and alike.
> > >
> > > We argue that the scope for this work is presenting a theoretical unsupervised method for relational learning and using illustrative experiments to highlight key properties/behaviors of the proposed method (read: not an application paper). While there are plenty of publicly available dataset (e.g. social networks, medical images, etc), they are not organized or posed as a relational learning task. For example, we think it is very interesting to apply relational learning to a pair of medical images from the same patient before/after treatment or diagnosis to extract relational properties irrespective of the patient (absolute property). While there are plenty of publicly available medical images from both before/after treatment or diagnosis, they are not ordered by patient id and any attempt to order these images will either require significant effort from domain experts or simply cannot be done due to ethical concern. When applying relational learning to unordered (by patient) image pairs, there is no clear separation of relational and absolute properties (this is also true for social network dataset). This in itself isn't necessarily a showstopper for us (as we do not need to know in advance the definition of relational&absolute property) but it does require significant effort to analyze/validate/interpret the results which we believe is outside the scope of this work. While we agree it is nice to include real-world examples, we believe they are not central to our current work and would reserve them for future study especially considering we already face challenges of including relevant information within the page limit.
> > >
> > > > Regarding other methods in the literature, there is a whole body of work in Statistical Relational Learning (SRL) with models like MLNs, DeepProbLog, ProbFOIL, SlipCover, PGMs etc or even works based on predicate invention and neurosymbolic approaches. Some of them are used for supervised learning, but, nevertheless, they learn new relations from primitive data or from other relations, and could be used for unsupervised learning.
> > >
> > > Please note that we have explicitly discussed Statistical Relational Learning (SRL) in our Introduction section where we have also included references to probabilistic logic programming (which the suggested methods are broadly based on). As you have correctly pointed out that all of the aforementioned methods require supervised (labelled) dataset for learning, we are very clear in our presentation that we are only focused on unsupervised learning. While we think it is conceivable to extent and augment supervised learning methods to a semi-supervised setting (learning from both labelled and unlabelled data), we are not aware of any supervised learning method that can be applied to a completely unsupervised setting. We would very much appreciate it if you can share more details regarding your suggestion on the usage of supervised method for unsupervised learning.
> > >
> > > > And regarding my question about the dates of your references? Aren't there any works more recent than 2019 in this area?
> > >
> > > Our priority is to include the most representative (not exhaustive) references that are immediately relevant to our discussion; the year of which they are published are not our main concern. For example, if a class of methods all share the same fundamental limitation in a relational learning context, we incline to only include the most representative work within that class of methods (either the original paper or a well-cited survey paper but not necessarily the most recent paper). Since our initial submission, we have included additional results from a recent self-supervised learning method BYOL (Grill et al., 2020) because they are immediately relevant to our problem settings.
> > > We welcome any suggestions that points us to other relevant work that we may have missed; meanwhile, we would gladly review our reference list.
> > >
> > > We hope this addressed all of your concerns.

---

> ### Author Response · Authors · 2021-11-29
> **Last day of final stage of discussion**
>
> Dear Reviewer Znwn:
>
> We are approaching the deadline of the final stage of discussion. At this time, we wanted to summarize our response to your outstanding concerns (please see our response below for a detailed discussion):
>
> 1. While we agree it is nice to include real-world examples, they are not central to our main message and it is appropriate to reserve them for future study as they are outside the scope of this work. Finally, while our examples are synthetic, the data itself are real-world images (MNIST, facial images).
>
> 2. We are very aware of related works from the field of Statistical Relational Learning (SRL) and have explicitly discussed them in our opening Introduction section. We included relevant references but did not go into details of each method as most of them do not fit our problem setting for obvious reasons. For examples, methods based on probabilistic logic programming (e.g., MLNs, DeepProbLog, ProbFOIL, SlipCover, etc.) typically require labelled data for training/learning (supervised learning methods),  whereas we only focus on a completely unsupervised setting in this work.
>
> 3. We hope you are convinced that we have included a comprehensive and representative list of references; meanwhile, we are more than happy to review/update our reference list.
>
> We would appreciated it if you can confirm that all your concerns have been addressed.

---

### Official Review · Reviewer_2C7q · 2021-11-02

**Correctness:** 3
**Technical Novelty And Significance:** 3
**Empirical Novelty And Significance:** 3
**Recommendation:** 6
**Confidence:** 3

**Main Review:**

I have 2 concerns regarding the presentation/organization of the paper:
1. The authors claim in the introduction that they formulate the problem of relational learning as a PGM which seems misleading. They approximate the problem because the omission of the independence condition is an important omission. In fact the authors themselves shed light on what could go wrong because of the omitted condition and use the RDPA trick to address it.
2. The relational learning problem needs better discussion. In particular, the transition from the definition of relational learning to the probabilistic formulation could use an example.

Other than this, I believe that this is an interesting paper with a novel formulation of the relational learning problem and extensive experimentation.

**Summary Of The Paper:**

In this paper the authors pose the problem of relational learning in the form of a Probabilistic graphical model and utilize variational inference for learning the relational property. The experiment section is extensive highlighting the utility of the proposed method.

**Summary Of The Review:**

Good paper help back by the presentation.

---

> ### Author Response · Authors · 2021-11-10
> **Author response to Reviewer 2C7q**
>
> Dear Reviewer 2C7q:
>
> Thank you for the detailed review and comments. Please find our response to your concerns/questions below:
>
> > The authors claim in the introduction that they formulate the problem of relational learning as a PGM which seems misleading. They approximate the problem because the omission of the independence condition is an important omission. In fact the authors themselves shed light on what could go wrong because of the omitted condition and use the RDPA trick to address it.
>
> We would appreciate if you can be more specific about which part of our claim you feel is misleading. Our current wording is "we encapsulate the relational learning problem with a PGM" which we believe is an accurate description. We clearly discussed the limitation of the proposed method in Sec 3 and Appendix B. We welcome any suggestion on revising our presentation or specific wording.
>
> > The relational learning problem needs better discussion. In particular, the transition from the definition of relational learning to the probabilistic formulation could use an example.
>
> We believe the motivation behind each condition in Eq. 1 as well as their connection to the relational learning definition is clearly stated in  Section 2. However, to provide additional explanation, we give the following well known example attributed to S. Bernstein (Hogg, 2005): Let $a$ and $b$ be two independent tosses of a fair coin, where we designate 1 for heads and 0 for tails. Let the third random variable $z$ be equal to 1 if exactly one of those coin tosses resulted in "heads", and 0 otherwise. It is easy to see that in this toy example $z$ satisfies all four conditions in Eq. 1 and we can intuitively interpret $z$ as meaning "different tosses" which is a relational property since it  indicates the relationship between $a$ and $b$ (same vs. different) and is independent of $a$ and $b$, i.e., observing $a$ or $b$ alone tells you nothing about $z$. The main difference between our definition in Eq. 1 and this toy example is that we do not specify what $z$ is. We only assume the existence of such $z$ and we want to learn about it. We hope the above example provides additional insight into our formulation and we would gladly add this example to our appendix if you find it useful.
>
> We hope this answers your question and look forward to additional discussion.

---

### Official Review · Reviewer_Jew5 · 2021-11-02

**Correctness:** 3
**Technical Novelty And Significance:** 3
**Empirical Novelty And Significance:** 3
**Recommendation:** 6
**Confidence:** 3

**Main Review:**

### Strength

- The relational learning problem is clearly formulated in the paper, and seems novel and interesting.
- The paper is well written and easy to follow.
- The experiments are well-designed and demonstrate interesting behavior of VRL.

### Questions/concerns

- In Table 1, why is STN-Rotate so much worse on Omniglot than on MNIST? Omniglot seems pretty similar to MNIST, and the two datasets are similarly constructed. Furthermore, STN-Rotate is essentially cheating (as it knows the transformations are rotations) so I would expect it to perform well. It is quite surprising that it does so badly on Omniglot.
- The comparison with STN seems a bit strange. Why does STN take both a and b as input? I would imagine an STN that takes just a as input and we compare the output with b. This is basically directly learning the transformations. In addition, for STN, clustering on the output of the trained localization network does not seem fair. Depending on how the STN is specified, I can imagine different outputs of the trained localization network that lead to the same transformation. Directly comparing the resulting transformations seems like the better approach.
- The VRL-PGM is not symmetric w.r.t. a and b. Does that affect the model in any way? Is this asymmetry considered in the training (e.g. randomizing the order of the two elements in each training example) and test process?
- For the Omniglot experiments, VRL with discrete no RPDA for 5 relationships is worse than VRL with continuous with no RPDA, so the claim that constraining z to be discrete improves performance does not always hold.
- How does the rotation augmentation work for some of the baseline methods, e.g. for VAEs? It seems by use of RPDA the model is cheating a bit, as it essentially has additional supervision signal coming from access to two pairs of samples that it knows share the same transformation. To make the comparison fair, for VAEs for example, in addition to training VAEs on the given training data, there should be additional supervision that the latent state for a pair of samples and the latent state for the pair of samples transformed by a relation preserving function should match. I don't know if this is the way rotation augmentation is used for VAEs. If not, it doesn't seem to be a fair comparison. Similar issues might exist for other baseline methods (although in terms of performance VAEs seem the most promising). If VAEs perform well with this additional piece of supervision information, then the good performance of VRL is more from the way RPDA is used, and less from the method itself.
- How does relational mapping look like on the Yale Face dataset? Some visualizations would be helpful. I expect it won't be able to generate transformed faces as that is quite complicated, but it would be interesting to see what the model really learned.


**Summary Of The Paper:**

This paper proposes an unsupervised learning method, variational relational learning (VRL), to address the relational learning problem and learn the underlying relationship between a pair of data irrespective of the nature of those data, and demonstrates its performance on variations of the MNIST/Omniglot datasets and the Yale Face datasets.

**Summary Of The Review:**

I have some questions/concerns about the paper so recommend weak reject, but would be happy to bump up the score if my questions/concerns can be satisfactorily addressed.

------------------------------
The authors have satisfactorily addressed my concerns, and I bumped up the score accordingly.

---

> ### Author Response · Authors · 2021-11-10
> **Author response to Reviewer Jew5**
>
> Dear Reviewer Jew5:
>
> Thank you for the detailed review and comments. Please find our response to your concerns/questions below (in the order they are raised):
>
> > why is STN-Rotate so much worse on Omniglot than on MNIST? ... STN-Rotate is essentially cheating (as it knows the transformations are rotations) so I would expect it to perform well.
>
> There are two observations that makes Omniglot task more challenging than MNIST: (1) Omniglot characters are more complex than MNIST hand written digits; (2) Unlike MNIST dataset, the Omniglot testing (hold-out) dataset consists of entirely different set of characters from its training dataset. We speculate the above observations combines with the fact that STN are trained to minimize mismatch between high-dimensional images cause STN-Rotate learning to frequently stuck on local minimum (this is partially reflected in the high variance of STN-Rotate clustering accuracy, especially for Omniglot 3-rel task). Our main message for presenting STN results is that the seemingly simple MNIST and Omniglot relational learning tasks are in fact very challenging even for specialized learning methods with STN-Rotate as an extreme example.
>
> > The comparison with STN seems a bit strange. Why does STN take both a and b as input? I would imagine an STN that takes just a as input and we compare the output with b.
>
> To clarify, the sampler in STN only takes $a$ as input and the goal is to transform it to $b$ using the output of localization net (and grid generator). The localization net require both $a$ and $b$ as input as it needs both piece of data to infer the relative relationship information.
>
> > clustering on the output of the trained localization network does not seem fair. Depending on how the STN is specified, I can imagine different outputs of the trained localization network that lead to the same transformation. Directly comparing the resulting transformations seems like the better approach.
>
> We used the same affine transformation function specified in the original STN paper and cluster on the output of the trained localization network because it provides all the information on the learned spatial transformation (e.g., relative rotation) between $a$ and $b$. In our controlled experiments, if the localization network produces different outputs that lead to the same transformation (e.g., +120 degree rotation vs. -240 degree rotation), then it must draw the distinction from the absolute property (digit representation, individual image orientation, etc.) and this goes against the goal of learning relative transformation independent from absolute property. Directly comparing the resulting transformations (we read as final transformed image $b$) would not be a good idea because different $a$ and spatial transformation (different relational property) can lead to the same $b$.
>
> > The VRL-PGM is not symmetric w.r.t. a and b. Does that affect the model in any way? Is this asymmetry considered in the training (e.g. randomizing the order of the two elements in each training example) and test process?
>
> This is a subtle but important question and we addressed this exact issue in Appendix A.1. In summary, in order to develop a tractable solution ,VRL-PGM is based a directed acyclic graph that artificially introduces conditional dependency between $a$ and $b$. We argued in Appendix A.1 that the application of VRL does not require the true conditional dependency between ($a$, $b$) be known in advance only that it is maintained consistently throughout learning and inference. In other word, VRL can be equally applied to either ($a$, $b$) or ($b$, $a$) as long as their position is maintained consistently. However, randomizing the order of the two elements will in general lead to a different problem, e.g., when we specifically wanted to learn undirected relationships as in our facial image experiments.
>
> > For the Omniglot experiments, VRL with discrete no RPDA for 5 relationships is worse than VRL with continuous with no RPDA, so the claim that constraining z to be discrete improves performance does not always hold.
>
> For the Omniglot 5-relationships tasks, we consider both VRL (Cat., no RPDA) and VRL (Cont., no RPDA) as completely failing the task; there is no meaningful distinction between clustering accuracy 22.4% vs. 32.1% (a random guess would have 20% accuracy). Instead, we focus on which how many tasks each method can solve: VRL (Cat., no RPDA) solves 2 out of 4 tasks and VRL (Cont., no RPDA) only solves one task.

---

> ### Author Response · Authors · 2021-11-10
> **Author response to Reviewer Jew5 - Cont.**
>
> > How does the rotation augmentation work for some of the baseline methods, e.g. for VAEs? It seems by use of RPDA the model is cheating a bit, as it essentially has additional supervision signal coming from access to two pairs of samples that it knows share the same transformation...
>
> First, all the methods we compared against in Table 1 uses conventional image rotation data augmentation during their training. Interestingly enough, we can (and have) apply RPDA to STN and NRI methods. However, we do agree with your assessment that RPDA may provide additional supervision signal coming from access to two pairs of samples that it knows share the same transformation. After our initial submission, we also wanted to test this idea and conducted another experiment based on a state-of-the-art contrastive self-supervised learning method BYOL (Grill et al., 2020). We think BYOL essentially captures your suggestion where it learns to reduce the distance between representations of different augmented views of the same image (‘positive pairs’) which we can easily generate using RPDA functions. Please find this additional experiment in our revised paper. In summary, BYOL also failed all the relational learning tasks for the same fundamental reason as other methods---there is no way to enforce the data representation learned by BYOL to be independent of absolute property. Note that the 'positive pairs' (two pairs of samples sharing the same transformation) generated by RPDA also shares most of their absolute properties (digit representation, etc.)
>
> > How does relational mapping look like on the Yale Face dataset? Some visualizations would be helpful. I expect it won't be able to generate transformed faces as that is quite complicated, but it would be interesting to see what the model really learned.
>
> As Yale Face dataset is extremely limited, the high-dimensional likelihood function completely overfits the data and we do not find the relational mapping results to be meaningful. However, the low-dimensional posterior function still is able to extract meaningful relational property.
>
> We hope this answers your question and look forward to additional discussion.

---

> ### Comment · Reviewer_Jew5 · 2021-11-18
> **Thanks for the response**
>
> I thank the authors for the detailed response.
>
> - Regarding STN on Omniglot: STN-Rotate performs very well on MNIST (which is understandable) so I wouldn't say the experiments demonstrate `the seemingly simple MNIST...relational learning tasks are in fact very challenging even for specialized learning methods with STN-Rotate as an extreme example.` Since STN-Rotate should be directly learning the rotation, I'm not sure the relative complexity of the two datasets matters much. So it might come down to Omniglot test datasets contain different characters. I think it would still be helpful to have a simple experiment with a customized train/test split, where the train and test data contain the same set of characters. If STN-Rotate still does not perform well I would like to understand better why that is the case.
>
> - Regarding RPDA for VAEs: the added results with BYOL seem much worse than those with VAEs. I don't feel this addresses my concerns. I would still like to see the experiment done with VAEs (which looks quite straightforward if regular VAE training is already in place), as that seems to be the most meaningful setup given the current set of results. This point about the use of RPDA is still a major unaddressed concern.
>
> - I do not expect the model to learn something amazing for the Yale face dataset. But I do think it would be informative to include some visualizations.

---

> > ### Author Response · Authors · 2021-11-19
> > **Author response to additional questions**
> >
> > Dear Reviewer Jew5:
> >
> > Thank you for your continuous attention and sharing with us your additional concerns. Please see our response to your questions:
> >
> > > Since STN-Rotate should be directly learning the rotation, I'm not sure the relative complexity of the two datasets matters much. So it might come down to Omniglot test datasets contain different characters. I think it would still be helpful to have a simple experiment with a customized train/test split, where the train and test data contain the same set of characters. If STN-Rotate still does not perform well I would like to understand better why that is the case.
> >
> > We repeated the experiment using your suggested customized train/test split and report that STN-Rotate performance does not improve. Based on our observation, the complexity of the data has significant impact on the learning performance. For example, we can improve STN-Rotate performance on Omniglot if we hand pick the training dataset to only contain simple characters that look distinctly different under image rotation. Another way to think about the effect of character complexity on STN-Rotate is that image mismatch (measured by BCE on pixels) can be significant for complex character underwent even small image rotation. This leads to our speculation that STN-Rotate frequently gets stuck in local minimum. To further support this claim, we report the clustering accuracy for individual runs of STN-Rotate on 3-relationships Omniglot tasks: (33.9, 33.4, 33.3, 99.5, 33.2 ). As can be seen, it is still possible (but not likely) for STN-Rotate to learn the relative relationships which is reflected in the high variance.
> >
> > >  I would still like to see the experiment done with VAEs (which looks quite straightforward if regular VAE training is already in place), as that seems to be the most meaningful setup given the current set of results. This point about the use of RPDA is still a major unaddressed concern.
> >
> > Based on your suggestion, we have conducted a new set of VAE experiments where we added an additional loss term to the VAE loss function that aim to minimize the difference between the latent representation from different RPDA augmented image pairs, i.e., $\alpha ||z - z'||_{2}$ where $z=Enc(a,b), z'=Enc(a',b')$ and we cross-validate the scalar $\alpha$ from 0.1 to 10. The clustering accuracy for the 4 relational learning tasks (cf. Table 1) are [92.9, 66.9, 99.0, 66.5]. Even with the added regularization on the latent space for VAE, there is no way to enforce the learned data representation be independent of absolute property since different RPDA augmented image still share large part of their absolute property (e.g., digit representation). For example, it is entirely possible that the modified VAE learns to tightly cluster images based on their digit representation (absolute property); this will lead to a small loss function value but makes the subsequent relational discrimination task more challenging since each digit representation can have all possible relative rotation relationships.
> >
> > >  I do not expect the model to learn something amazing for the Yale face dataset. But I do think it would be informative to include some visualizations.
> >
> > Please see the rebuttal revision for our newly added relational mapping examples for Yale dataset in Figure 11 in Appendix E.5.
> >
> > We hope this address all your questions and concerns.

---

> > > ### Comment · Reviewer_Jew5 · 2021-11-20
> > > **Thanks for the response**
> > >
> > > I thank the authors for the additional experiments and for addressing my concerns. The results with STN-Rotate on Omniglot are interesting, as it seems to suggest STN-Rotate has trouble learning invariance for complex digits. The results with VAEs make sense. I have bumped up my score accordingly.

---

> > > > ### Author Response · Authors · 2021-11-20
> > > > **Thank you for reevaluating our work**
> > > >
> > > > Dear Reviewer Jew5:
> > > >
> > > > Thank you for your continuous interest and reevaluate our work. We thoroughly enjoyed the discussion!

---

### Official Review · Reviewer_ahUC · 2021-11-02

**Correctness:** 3
**Technical Novelty And Significance:** 3
**Empirical Novelty And Significance:** 2
**Recommendation:** 6
**Confidence:** 3

**Main Review:**

A new relational learning approach is proposed based on variational learning. The main idea is to disentangle absolute and relational properties and force the learner to learn based on relationships rather than absolute properties of instances. The problem is formulated as a variational inference problem and a data augmentation method is developed that preserves relational information. Experiments are shown on MNIST, monoglot and Yale face database for unsupervised learning.

The paper has an interesting idea and I liked the formulation of the variational inference problem for relational learning. The main weakness is perhaps the somewhat forced construction of relationships in the empirical results. Since the paper has a very general aim of learning relationships, it would have been really strong if this could be shown through a natural relational problem rather than the constructed ones. The results do show that the method performs better than even a specialized transformer model for rotations (STN) which is definitely.a plus. On the other hand, the RPDA functions need to encode the rotational aspects so it does require some specialization. I was not sure about the data augmentation RPDA functions for the facial emotions data (was it related to illumination changes, etc.).  Also, how different is the relational data augmentation from the typical types of data augmentation methods that people would use if they had the domain knowledge, maybe the speciality of the relational augmentation methods could be highlighted a bit better.

After author feedback

I thank the authors for their feedback. The idea definitely seems good, maybe with a bit stronger comparison/ empirical studies, this would be a stronger paper.

**Summary Of The Paper:**

A new approach for relational learning is proposed. The main contributions are
A novel formulation of relational learning as a variational inference problem
A novel augmentation method for relational data
Empirical results that show the ability to learn relationships in images.

**Summary Of The Review:**

Overall the formulation seems interesting and results show that the general approach is on par with state of the art specialized methods for the tasks considered. However, the relationships in the experiments seem a bit forced and maybe simpler data augmentation methods could replace the relational augmentation methods used.

---

> ### Author Response · Authors · 2021-11-09
> **Author response to Reviewer ahUC**
>
> Dear Reviewer ahUC:
>
> Thank you for the detailed review and comments. Please find our response to your questions below:
>
> > The main weakness is perhaps the somewhat forced construction of relationships in the empirical results. Since the paper has a very general aim of learning relationships, it would have been really strong if this could be shown through a natural relational problem rather than the constructed ones.
>
> This is a very good suggestion but we believe this is outside the scope of this current work for the following reasons. There is no existing benchmark relational learning tasks for unsupervised methods and most natural relational problems require significant effort from domain expert to build a large-scale training dataset. Another problem with conventional relational problems is that most of them rely on subjective interpretation that can only generate anecdotal evidence and not conducive for rigorous quantitative evaluation. As an initial investigation into this overlooked problem, our goal is to design a controlled experiments that allow us to rigorously and quantitatively test various relational learning tasks as well as providing a common problem settings where we can compare with a wide range of existing unsupervised methods. We believe our empirical results achieved these goals. We do believe our method can be applied to a much wider problem setting as well as other data modality after proper adjustment to likelihood and posterior definition; however, we propose to pursue this in a future work.
>
> > the RPDA functions need to encode the rotational aspects so it does require some specialization.
>
> The proposed RPDA is based on the same principle as conventional data augmentation method: the learning task is assumed to be invariant to the adopted data augmentation functions. And just like any data augmentation, the effectiveness of RPDA will depend on the problem setting and we advocate to start without RPDA and only apply it when suspecting information-shortcut occurs. This is discussed in Appendix A.2.
>
> > I was not sure about the data augmentation RPDA functions for the facial emotions data (was it related to illumination changes, etc.)
>
> Figure 5 and 6 are two separate and independent experiments. Figure 5 only contain emotional change facial images and Figure 6 only contain illumination change facial images. Both experiments uses the same VRL model and RPDA functions: image rotation and swapping. One does not depend on the other.
>
> > how different is the relational data augmentation from the typical types of data augmentation methods that people would use if they had the domain knowledge
>
> We believe the main contribution of RPDA is not "what" data augmentation functions are used (e.g., image rotation, etc.) but "how" they are used which is detailed in Eq. (4). In other words, if tailored data augmentation functions can be devised based on domain knowledge, those data augmentation functions can be easily incorporated into RPDA by following Eq. (4).
>
> > maybe the speciality of the relational augmentation methods could be highlighted a bit better.
>
> Due to the limited space, a detailed discussion on the practical applicability of RPDA is presented in Appendix A.2
>
> >  the relationships in the experiments seem a bit forced and maybe simpler data augmentation methods could replace the relational augmentation methods used.
>
> All the methods we compared against in Table 1 uses conventional image rotation data augmentation during their training. Furthermore, in our ablation study in Table 1, VRL (Cont., no RPDA) uses the same image rotation data augmentation in the conventional way and not following Eq. (4). These results clearly show that conventional data augmentation alone is not enough to solve the relational learning problem.
>
> We hope this answers your question and look forward to additional discussion.

---

### Official Review · Reviewer_7vpw · 2021-11-03

**Correctness:** 3
**Technical Novelty And Significance:** 3
**Empirical Novelty And Significance:** 3
**Recommendation:** 6
**Confidence:** 3

**Main Review:**

Strengths:

The problem set up is novel, and the paper addresses an important problem. The method is intuitive, and the RPDA is novel. The experiment is well-conducted, and shows that the method outperforms other strong baselines.

Weaknesses:
In Eq. 1, the authors introduces 4 conditions for z to be satisfied as relational property. However, the proposed PGM only satisfies 3 of them where (ii) is not satisfied. This is an important omission. The authors may need to consider ways to make it consistent, or justify such omission.

--
Update:
I have read the reply from the author as well as other reviewers' comments. It is understandable that the 4 conditions are hard to achieve simultaneously, and I'm satisfied with the authors' response. Considering the weaknesses of the experiments as raised by other reviewers, I remain my score of borderline accept.

**Summary Of The Paper:**

This paper introduces a variational method for relational learning. It first introduces relational learning as learning based on relational property instead of absolute property, and introduces conditions (Eq. 1). Then it proposes VRL-PGM with a variational lower bound. To eliminate the information short-cut, it introduces relation-preserving data augmentation (RPDA). Experiments show that the method is able to perform relation discrimination and relation mapping, on a variant of MNIST and Yale face datasets.

**Summary Of The Review:**

In summary, the authors addresses an interesting problem and proposes a novel method, and overall an interesting paper. The authors may need to improve its justification of the PGM omitting condition (ii), or find ways to make it consistent with the proposed 4 conditions of relational variable.

---

> ### Author Response · Authors · 2021-11-09
> **Author response to Reviewer 7vpw**
>
> Dear Reviewer 7vpw:
>
> Thank you for the detailed review and comments. Please find our response to your questions below:
>
> > Weaknesses: In Eq. 1, the authors introduces 4 conditions for z to be satisfied as relational property. However, the proposed PGM only satisfies 3 of them where (ii) is not satisfied. This is an important omission. The authors may need to consider ways to make it consistent, or justify such omission.
>
> We addressed this exact issue at the beginning of Section 3 where we explained that it is very difficult (to our knowledge) to develop a tractable learning method that can simultaneously satisfy all 4 conditions in Eq. 1. As a compromise, we propose VRL in Section 3.1 to satisfy 3 (out of 4) constraints in exchange for a tractable learning method. However, as you pointed out, we cannot just ignore Eq. 1(ii) so we presented VRL’s unique optimization challenges in Section 3.2 that are directly attributable to the omission of Eq. 1(ii) and proposed several mitigation strategy (e.g., RPDA) to help avoid some worst-case learning scenarios. Additional discussion is presented in Appendix A.1 where we further delve into the implication of VRL-PGM on relational learning. Our goal (and contribution) of introducing Eq. 1 is to propose a probabilistic formulation for the relational learning problem; even if we did not completely solve Eq. 1, we believe it is important to put it forward to advance our understanding of the problem especially in the unsupervised learning context.
>
> In summary, satisfying all four conditions in Eq. 1 (especially (i) and (ii)) is hard. We believe the main contribution of VRL is a tractable solution that guarantees to satisfy 3 (out of 4) conditions in Eq. 1 and provides additional safeguards (e.g. RPDA) that avoids worst-case learning scenarios that are caused by the omission of Eq.1 (ii).
>
> We hope this answers your question and look forward to additional discussion.

---

### Author Response · Authors · 2021-11-26
**Final stage discussion period: last call for questions**

Dear Reviewers:

We thank you again for your time and review. We took your questions very serious and made our best effort to address your concerns in a timely manner. At this time we would like to kindly ask you to review our response and let us know if you have additional questions. We welcome any feedbacks and look forward to hear back from you. However, if you do find our response satisfactory, we would greatly appreciate it if you could take our response into consideration and reevaluate our work. Thank you!!

---

### Public Comment · ~Kuang-Hung_Liu1 · 2022-02-04
**Summary of camera ready revision changes**

We thank anonymous reviewers, AC, and PC for feedbacks that helped improve the paper and the opportunity to present our work. We have incorporate all the feedbacks and prepared a camera ready revision with the following summary changes:

1. We added a motivating example for Eq. 1 in the appendix as suggested by Reviewer 2C7q.

2. We added a detailed discussion with statistical relational learning methods in Related Work. This also include our discussion with Reviewer Znwn.

3. We added an additional baseline method comparison---denoted as VAE-contrastive---in our Table 1 as suggested by Reviewer Jew5. Although we could not find a reference for the exact method, we think the work by [Chen et al., (2020) A simple framework for contrastive learning of visual representations] is similar in concept.

4. We respond to Reviewer Znwn's comments that our experiments lacks diversity (only use image data) by adding an additional experiment in Section 5.3 that shows the learning of relative speech (time-series waveform) emotion changes.

5. We provided link to source code for reproducible experimental results (https://github.com/kh1iu/vrl.git).

6. To accommodate the above changes, we move parts of original main text to appendix.

7. We have updated our reference list as suggested by Reviewer Znwn.

We welcome any additional comments and suggestions. Thank you!

---

### Decision · Program_Chairs · 2022-01-20

**Decision:**

Accept (Poster)

**Comment:**

This papers studies the classical problem of relational learning from a probabilistic perspective. The authors propose four reasonable constraints to encode relational properties, and develop a PGM-based variational method for learning relational properties from data. After extensive discussion with the authors, a majority of the reviewer reviewers agree the approach is interesting, if not without some flaws.

The problem studied is interesting, novel, and could lead to new developments in the area of relational learning. It is expected that the experiments have some limitations given the authors have approached the problem from a fresh new angle, which the reviewers have appreciated.

Please pay attention to the suggestions from the reviewers, and in particular, please add a more detailed discussion with statistical relational learning: This material may not be familiar to the broader ML audience, and therefore it is essential to make these comparisons explicit.